# Machine learning inference of continuous single-cell state transitions during myoblast differentiation and fusion

Amit Shakarchy[1,4], Giulia Zarfati[2,4], Adi Hazak [ID][2,4], Reut Mealem[1], Karina Huk [ID][2], Tamar Ziv [ID][3], Ori Avinoam [ID][2✉] & Assaf Zaritsky [ID][1✉]

## Abstract

Cells modify their internal organization during continuous state transitions, supporting functions from cell division to differentiation. However, tools to measure dynamic physiological states of individual transitioning cells are lacking. We combined live-cell imaging and machine learning to monitor ERK1/2-inhibited primary murine skeletal muscle precursor cells, that transition rapidly and robustly from proliferating myoblasts to post-mitotic myocytes and then fuse, forming multinucleated myotubes. Our models, trained using motility or actin intensity features from single-cell tracking data, effectively tracked real-time continuous differentiation, revealing that differentiation occurs 7.5–14.5 h post induction, followed by fusion ~3 h later. Co-inhibition of ERK1/2 and p38 led to differentiation without fusion. Our model inferred co-inhibition leads to terminal differentiation, indicating that p38 is specifically required for transitioning from terminal differentiation to fusion. Our model also predicted that co-inhibition leads to changes in actin dynamics. Mass spectrometry supported these in silico predictions and suggested novel fusion and maturation regulators downstream of differentiation. Collectively, this approach can be adapted to various biological processes to uncover novel links between dynamic single-cell states and their functional outcomes.

**Key words** State Transition; Machine Learning; Myogenesis; Myoblast Fusion; Differentiation
**Subject Categories** Computational Biology; Development; Methods & Resources

## Introduction

Single-cell transitions via dynamic changes in protein expression, intracellular organization, morphology, and function, drive many important biological processes, such as the progression through the phases of the cell cycle, cellular differentiation, the transition from an immotile to a motile state, or from a living to an apoptotic state. Aberrant cell-state transitions lead to various diseases, including cancer and neuromuscular disorders. As such, single-cell state transitions play an inherent role in physiological processes such as embryonic development, tissue regeneration, and in various pathologies.

Obtaining a holistic mechanistic understanding of these processes relies on the ability to continuously measure the physiological state of a cell through time. However, technical limitations, such as the number of live fluorescent state transition reporters that can be simultaneously imaged, hinder the elucidation of cell-state transitions as continuous processes. Moreover, state markers that could provide a continuous description are unknown for many biological processes. Consequently, we are currently limited to studying discrete cell states with missing intermediate states, which are often critical (Stumpf et al, 2017; Szkalisity et al, 2021).

Attempts to quantitatively follow cell-state dynamics have focused on the computational construction of "pseudo-time" trajectories from the integration of fixed cell images (Eulenberg et al, 2017; Gut et al, 2015; Rappez et al, 2020; Stallaert et al, 2022; Szkalisity et al, 2021; Yang et al, 2020). However, the capacity to identify single-cell trajectories that deviate from the most common progression, is limited in this approach, due to heterogeneity (Schroeder, 2011). Live-cell imaging offers a solution to this challenge by enabling dynamic monitoring and extraction of temporal information at the single-cell resolution, with the caveat that unsupervised modeling may consider extrinsic factors that are unrelated to the state transition, which may confound proper modeling of the continuous process (Copperman et al, 2021; Wang et al, 2022; Wu et al, 2022).

The formation of multinucleated muscle fibers is an essential state transition for vertebrate muscle development and regeneration. Following injury or growth stimuli, quiescent muscle progenitors called Satellite cells become activated to augment the muscle. At the onset of this process, activated satellite cells (myoblasts) express myogenic regulatory factors such as MyoD and proliferate to generate the myogenic progenitors needed for muscle

---

[1]Department of Software and Information Systems Engineering, Ben-Gurion University of the Negev, Beer-Sheva 84105, Israel. [2]Department of Biomolecular Sciences, Weizmann Institute of Science, Rehovot 761001, Israel. [3]The Smoler Proteomics Center, Lorry I. Lokey Interdisciplinary Center for Life Sciences and Engineering, Technion Israel Institute of Technology, Haifa 3200003, Israel. [4]These authors contributed equally: Amit Shakarchy, Giulia Zarfati, Adi Hazak. ✉E-mail: ori.avinoam@weizmann.ac.il; assafzar@gmail.com

regeneration (Bischoff, 1986; Hurme and Kalimo, 1992; Schmidt et al, 2019). Next, myoblasts upregulate the expression of factors such as Myogenin (MyoG) to exit the cell cycle and initiate terminal differentiation (Hernández-Hernández et al, 2017; Lepper et al, 2011; Singh and Dilworth, 2013). Myoblasts initially differentiate into elongated fusion-competent myocytes that migrate, adhere, and fuse with the regenerating muscle fibers (Abmayr and Pavlath, 2012). Newly formed myofibers are characterized by the expression of myosin heavy chain (MyHC; (Bentzinger et al, 2012; Lepper et al, 2011; Yin et al, 2013).

Although significant progress has been made in understanding muscle development, myoblast differentiation and fusion, remain incompletely understood at the molecular and cellular levels owing to several technical challenges. First, myoblasts differentiation and fusion are complex heterogeneous events, confounding systematic investigation. Second, while proliferating and terminally differentiated cells are relatively easy to distinguish morphologically, there are no markers for intermediate stages of differentiation or means for correlating between differentiation state and specific markers or functions such as motility, morphology, or signaling.

In this study, we combined live-cell imaging with supervised machine learning to quantitatively monitor the differentiation state of individual skeletal muscle precursor cells as they differentiate and fuse to form multinucleated muscle fibers ex vivo. Leveraging single-cell trajectories from time-series data, we trained machine-learning models to evaluate the differentiation state of these cells. Specifically, differentiation was initiated by pharmacologically inhibiting extracellular signal-regulated kinases (ERK1/2), leading to differentiation and myotube formation within 18–24 h (Eigler et al, 2021). Our models, which were trained using cell motility and/or actin intensity time-series-derived features, distinguished between proliferating myoblasts and terminally differentiated myocytes. We hypothesized that the models' score can serve as a continuous readout for the cells' differentiation state and proceeded to validate this hypothesis and explore its potential for uncovering novel biological insights. We found that the average differentiation score maintained a steady increase between 7.5 and 14.5 h post induction, mirroring the continuous state transition from post-mitotic undifferentiated myoblasts to terminally differentiated myocytes. In addition, the predicted differentiation state at the single-cell level correlated with the time of fusion, suggesting that differentiation and fusion are sequential and coordinated yet distinct processes.

Building on this foundation, we employed a combined pharmacological approach, inhibiting both ERK1/2 and p38. This led to an apparent accumulation of terminally differentiated, yet unfused cells. Our model deduced that these cells have transitioned to terminal differentiation and implied that co-inhibition leads to changes in actin dynamics. This inference by our model was experimentally validated using both immunofluorescence and mass spectrometry. Mass spectrometry also revealed a group of proteins significantly downregulated in co-inhibited unfused cells compared to those differentiated by ERK1/2 inhibition. Notably, the expression of the fusion-specific protein, Myomixer (Zhang et al, 2017; Quinn et al, 2017; Bi et al, 2017) was abolished in co-inhibited cells. In addition, several actin regulators were upregulated in ERK1/2 inhibited compared to cell where both ERK1/2 and p38 were inhibited, confirming our model's prediction regarding

alteration in the actin machinery. Our findings demonstrate the potential to mechanistically uncouple differentiation from fusion, emphasizing the pivotal role of p38 in orchestrating the transition from differentiation to fusion, and revealing potential novel regulators of the late differentiation and fusion. Collectively, our findings demonstrate a method to quantify cellular state transitions that can be adapted to other continuous processes.

# Results

## Differentiation correlates with reduced motility and increased actin intensity

We previously established that ERK1/2 inhibition induced robust, faster, and less temporally variable differentiation and fusion of primary myoblasts isolated from chick and mice, leading to the rapid formation of myotubes ex vivo within 24 h post induction (Eigler et al, 2021). Immunofluorescence staining of the differentiation markers MyoG and MyHC at different time points in cultures treated with the ERK1/2 inhibitor SCH 772984 (ERKi, 1 μM) or with DMSO as control (Morris et al, 2013) show that differentiation into fusion-competent myocytes is accompanied by the upregulation of MyoG, which initiates terminal differentiation (Figs. 1A and EV1 and EV2). MyHC expression begins 16 h post induction and peaks in the multinucleated myotubes at 24 h (Figs. 1A and EV2). The number of MyoG-expressing cells increases over time, stabilizing at 14 h post induction (Figs. 1B and EV1 and EV2).

To characterize the dynamic behavior of differentiating myoblasts, we isolated primary myoblasts from mice co-expressing the nuclear marker tdTomato fused to a nuclear localization signal (tDTomato-NLS) (Prigge et al, 2013) and the F-actin marker LifeAct-EGFP (Riedl et al, 2010; Fig. 1C). Actin governs a range of processes during differentiation, from cell motility, elongation, and alignment to fusion and fibrillogenesis, ultimately leading to sarcomere formation post-fusion and the subsequent maturation of myotubes into myofibers (Rubinstein et al, 1976; Otey et al, 1988; Nowak et al, 2009; Luo et al, 2022).

To collect dynamic information on the continuous transition from proliferation to terminal differentiation we performed time-lapse widefield microscopy of large fields of view each containing ~3000 cells (Fig. 1C; Movie EV1). Cultures were imaged for 23 h, starting 1.5 h after ERKi or DMSO (control) treatment. We observed that differentiation was accompanied by a decrease in cell motility, consistent with previous studies showing that myocytes are less motile than myoblasts ex vivo (Powell, 1973; Griffin et al, 2010) (Fig. 1D; Movie EV1). The regulated reduction in cellular motility is beneficial for enhancing the cell–cell interactions that would initiate differentiation and subsequently promote fusion (Krauss et al, 2005; Buggenthin et al, 2017; Nowak et al, 2009; Luo et al, 2022). Similarly, we observed an increase in the fluorescence intensity of the F-actin marker most likely corresponding to the expression of muscle-specific actin isoforms (Fig. 1C,E). Cumulatively, these experiments demonstrate that the transition of myoblasts from the proliferative to the terminally differentiated state is accompanied by dynamic changes in motility and actin intensity.

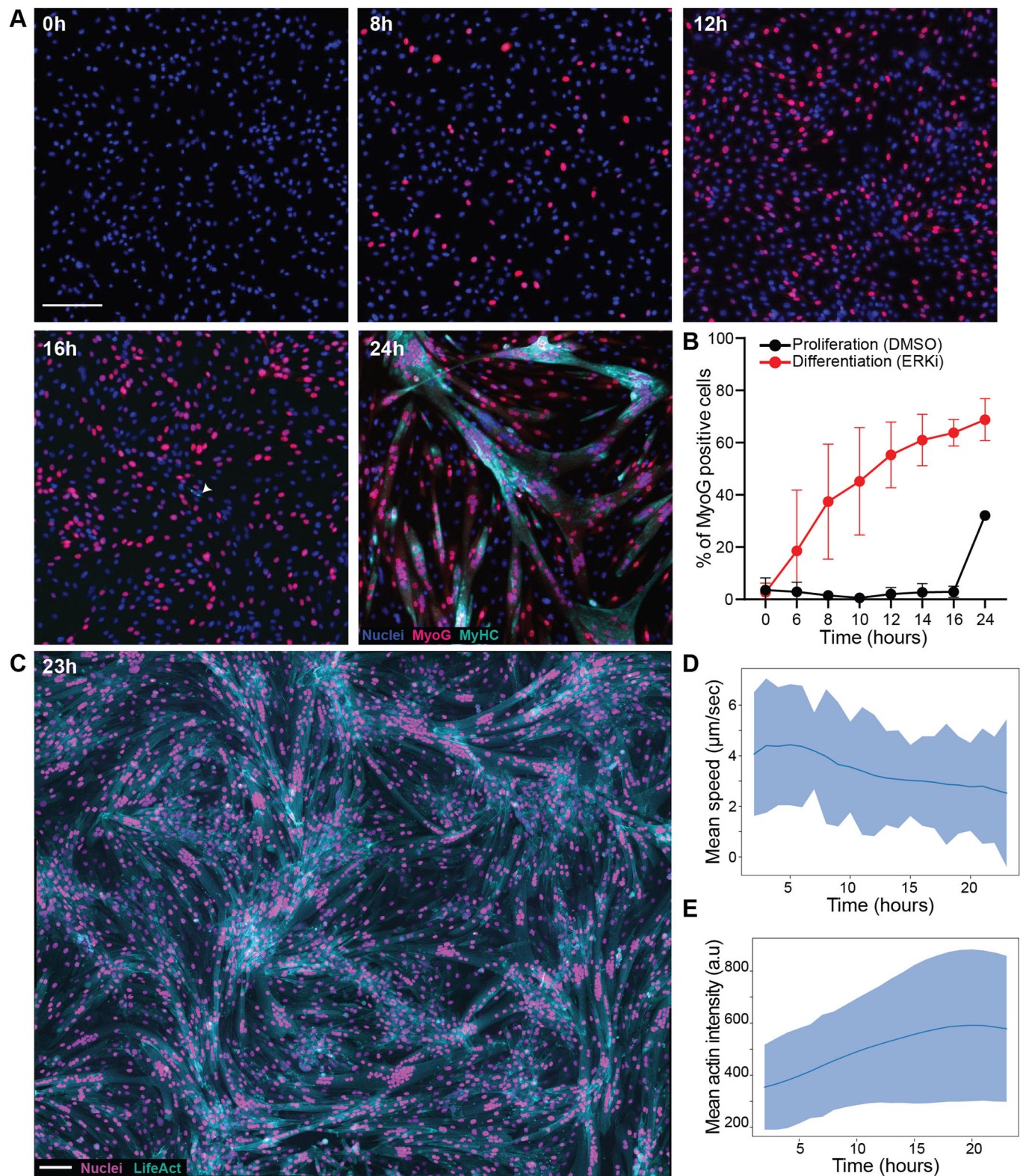

**B** — Proliferation (DMSO), Differentiation (ERKi)

**D** Mean speed (μm/sec) vs Time (hours)

**E** Mean actin intensity (a.u) vs Time (hours)

## Machine learning applied to time-series data generates quantitative single-cell differentiation trajectories

Following the association between differentiation and the population scale changes in actin intensity and motility, we hypothesized that the information encoded in single-cell migration trajectories and actin dynamics might be sufficient to computationally estimate a continuous score reflecting a myoblast's gradual transition from an undifferentiated proliferative state to a terminally differentiated fusion-competent state. To test this hypothesis, we took a machine-

**Figure 1. Myoblast differentiation ex vivo leads to dynamic changes in motility and actin intensity.**

(A) Overlay images of primary myoblasts fixed at different time points after ERK inhibition and stained for MyoG (red), MyHC (green), and the nuclei (Hoechst, blue). Magnification ×5. Scale bar: 100 μm. (B) Percentage of MyoG-positive cells in differentiating cultures (ERKi, red) and proliferating controls (DMSO, black) over time. The error bar represents the standard deviation (SD) of the mean (n >50,000 cells per condition). (C) Overlay image of primary myoblasts expressing the nuclear marker tDTomatto-NLS (magenta) and the actin marker LifeAct-EGFP (cyan) 23h after ERKi treatment. Magnification ×10. Scale bars 100 μm. (see Movie EV1). (D) Mean (line) and standard deviation (shade) of single-cell speed over time during differentiation (ERKi; ~3000 cells). (E) Mean (line) and standard deviation (shade) of actin intensity over time of an entire field of view of differentiating cells (ERKi; "Methods").

learning approach: (1) extracting features from the motility/actin time series, (2) training machine-learning classification models (aka *classifiers*) to discriminate between the undifferentiated and differentiated states, and (3) using the confidence of these models as a quantitative measurement for cell state.

The first step in designing our machine-learning model was determining which cells and time frame can be considered as undifferentiated or differentiated. Myocytes must differentiate to become fusion-competent (Abmayr and Pavlath, 2012). Hence, we defined the cultures as differentiated for classification 2.5 h before the first fusion event was observed in the field of view. Undifferentiated cells were defined from the population grown in proliferation medium in the presence of DMSO, which continue to proliferate and remain undifferentiated, except a small fraction that begins to differentiate stochastically toward the end of the experiment due to the increase in cell density (Eigler et al, 2021). To enable continuous scoring along single-cell differentiation trajectories, we performed semi-manual single-cell tracking, where each trajectory was manually verified and corrected when necessary (Movie EV2). Single-cell analysis confirmed our population-based results that differentiation was accompanied by a decrease in cell motility and an increase in the F-actin marker's fluorescence intensity (Appendix Fig. S1). We partitioned trajectories of undifferentiated and differentiated myoblasts to overlapping temporal segments of 2.5 h each, for an overall 16,636 undiffer-entiated and 47,819 differentiated temporal segments, extracted from 310 and 538 cells correspondingly, that were used for model training (Fig. 2A, top). From each temporal segment, we extracted the corresponding single-cell motility (dx/dt, dy/dt) and actin intensity time series. Single-cell motility/actin time-series features were extracted using the Python package "Time Series FeatuRe Extraction on the basis of Scalable Hypothesis tests" (tsfresh) that derives properties such as temporal peaks, descriptive statistics (e.g., mean, standard deviation) and autocorrelations (Christ et al, 2018). The extracted single-cell feature vectors and their corre-sponding undifferentiated/differentiated labels were used to train random forest classifiers (Breiman, 2001), which surpassed other machine-learning algorithms (Appendix Fig. S2). The entire process is depicted in Fig. 2A and detailed in "Methods".

We applied the trained motility and actin classifiers on single-cell trajectories from an experiment that was not used for training and attained a continuous quantification following the differentia-tion process by using overlapping temporal segments. At the population level, the single-cell-state classification performance gradually increased from an area under the receiver operating characteristic (ROC) curve (AUC) of ~0.6 to ~0.85 at 7.5–14.5 h from experimental onset (Fig. 2B,C). A classifier trained on features derived from both motility and actin time series surpassed each of the motility/actin classifiers, suggesting that motility and actin dynamics contain complementary information regarding the cells'

state (Fig. EV3A–C). The AUC values of all models were well beyond the random value of 0.5, indicating that our classifiers can discriminate between undifferentiated and differentiated cells at the population level before appreciable cell morphological changes occur.

Next, we wondered whether we can use these classifiers to predict the differentiation state of a single cell. For a given temporal segment of a given cell, the classifier outputs a "confidence score" (i.e., differentiation score) that reflects the model's certainty in its prediction. Low differentiation scores indicate that the cells are predicted as undifferentiated, while high scores indicate predicted differentiation. To interpret what temporal features were the most important for the model's prediction, we applied SHapley Additive exPlanations (SHAP) (Lundberg and Lee, 2017) and used random forest's feature importance algorithms (Breiman, 2001). Both interpretability methods highlighted temporal features related to high variance of acceleration rate or high complexity of actin intensity time series as dominant features driving the models' prediction (Appendix Fig. S3). We hypothesized that the differ-entiation score could be used as a continuous readout for the cell state. At the critical time frame of 7.5–14.5 h, at the population level, the differentiation scores of ERKi-treated cells gradually increased for the motility (Fig. 2D), the actin-based (Fig. 2E), and the combined (Fig. EV3D,E) models while maintaining low scores for experiments of DMSO-treated cells. For the rest of the manuscript, we focused on analyzing the motility- and actin-based models, because showing that each of two independent models trained with different temporal readouts (motility, actin) can quantitatively monitor the cell differentiation process, and thus strengthening our goal of evaluating whether the confidence score of a classifier trained for a binary classification task can be used to continuously measure a biological process that evolves over time.

We conducted a single-cell analysis by measuring the Spearman correlation between single-cell differentiation score and time at the critical time interval of 7.5–14.5 h when differentiation occurs. This analysis indicated that most cells underwent a monotonic increase in differentiation scores over time (Fig. 2F). A similar gradual increase in differentiation score at 7.5–14.5 was observed when flipping the experiments used for training and testing (Appendix Fig. S4), the differentiation score was not sensitive to the size of the temporal segment (Appendix Fig. S5), nor to the window size used to measure actin (Appendix Fig. S6), and was consistent across multiple independent trainings (Appendix Fig. S7). Visualizing single-cell trajectories showed that most trajectories followed a gradual increase in their differentiation scores (Fig. 2G). Measuring the distribution of the per-cell differentiation scores' temporal derivative (Appendix Fig. S8A,B), their integration over time (Appendix Fig. S8C–F), the predicted onset (Appendix Fig. S9), and the predicted duration (Fig. 2H) of the differentiation process, suggested that the progression in single-cell differentiation is highly

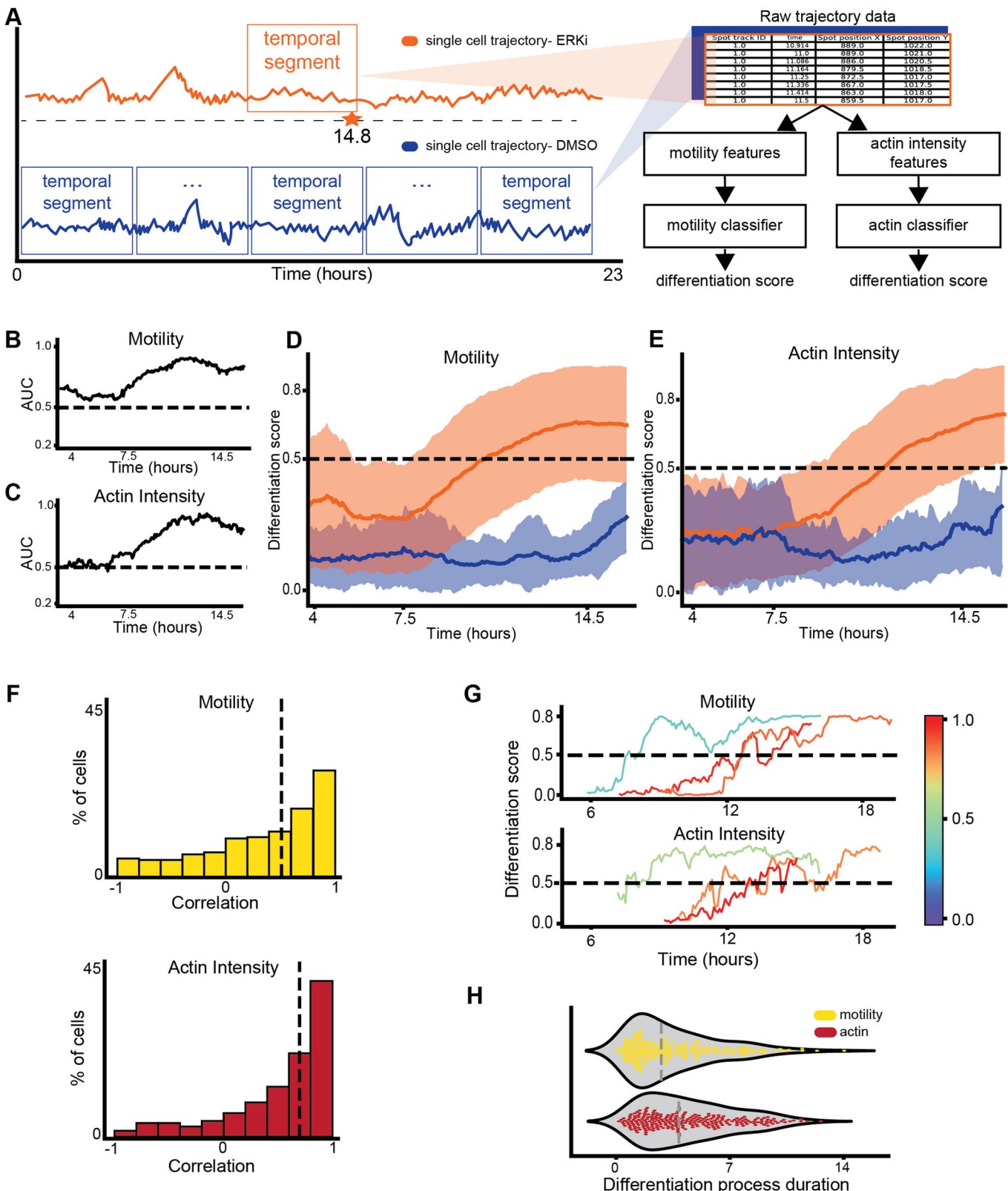

**Figure 2.  Inference of single-cell differentiation trajectories by machine learning applied to actin/motility dynamics.**

(**A**) Training Random Forest classifiers to predict single cells' differentiation state—cartoon. Left: single-cell motility/actin time series are partitioned into temporal segments of 2.5 h each. Positive labels were assigned to the ERKi-treated cells' segment (top, orange) starting 2.5 h before the first fusion event (orange star on the dashed timeline). Negative labels were assigned to all segments of DMSO-treated cells (blue). Right: features extracted from the positive (orange) and negative (blue) time series (top) were used to train classification models. Two models, one based on motility and the other on actin intensity, were trained based on time series extracted from the single-cell trajectories. (**B, C**) Classification performance. Area under the receiver operating characteristic (ROC) curve (AUC) over time for classifiers trained with motility (**B**) and actin intensity (**C**) time series. The AUC was calculated for 789 cells from an independent experiment. The classification performance of a random model (AUC = 0.5) is marked with a dashed horizontal line. (**D, E**) Mean (solid line) and standard deviation (shade) of the differentiation score over time of ERKi (orange) and DMSO (blue) treated cells using the motility (**D**) and the actin intensity (**E**) classifiers (ERK: 575 cells; DMSO: 103 cells). The analysis for the entire experiment is shown at (Appendix Fig. S17). (**F**) Distribution of single-cell Spearman correlation between the classifier's score and time, calculated for motility (orange) and actin (red) classifiers (N = 575 cells). Median correlation coefficient values were 0.55 (motility) and 0.7 (actin). In total, 66.1% (motility) and 73.4% (actin) of the cells showed a significant correlation (P value < 0.05), as assessed by Spearman's rank-order correlation test. (**G**) Representative single cells differentiation trajectories inferred by the motility (top) and the actin (bottom) classifiers. Trajectories are colored according to the Spearman correlation between their corresponding differentiation score and time. (**H**) Distribution of the single cell predicted duration of the differentiation process, as measured by the motility (yellow) and actin intensity (red) classifiers' prediction: the time passed between a stable low threshold of 0.2–0.3 and a high stable threshold of 0.7–0.8 (full details in "Methods"). The median predicted duration of the differentiation process was 3.3 (motility) and 4.5 (actin intensity) hours (N = 575 cells).

heterogeneous (Fig. 2H). These results suggest a heterogeneous gradual transition from an undifferentiated to a differentiated state within a typical time frame.

The motility- and actin-based classifiers' predictions were mostly consistent at their single-cell predictions, showing high correlation beyond 0.5 for 62% of cells, and negative correlation for less than 4%, and providing further support that both models measure the continuous state transition (Appendix Fig. S10A,B). Analysis of cells sub-groups partitioned according to the agreement between the motility and actin models, showed that lower agreement between the classifiers was associated with lower monotonicity of the differentiation scores of both models (Appendix Fig. S10C). In most cases of disagreement between the motility and actin models, we noticed a deviation in the actin-based model when the cell entered a crowded region and/or crawled below other cells (Appendix Fig. S11). High agreement between the two models was associated with the single cells' motility persistence, the ratio between the direct translation (i.e., distance from start to end), and the overall distance traveled (Appendix Fig. S10C). Qualitative and quantitative association between single cells' motility persistence and motility-based (but not actin-based) differentiation score identified persistent motility as a functional marker for the intermediate states of myoblast differentiation (Fig. EV4). The lack of association between the actin model and persistence suggests that the actin model encodes different dynamic properties that are linked to the differentiation. Moreover, this lack of association highlights the potential to use deviations between these models to discover mechanisms that uncouple the link between motility, actin dynamics, and myoblast differentiation. Altogether, our data suggest that machine learning can transform motility and actin dynamics to a quantitative readout characterizing the myoblast differentiation process at single-cell resolution describing a continuous myoblast state transition from an undifferentiated to the terminally differentiated states at 7.5–14.5 h post induction.

## Models that discriminate between undifferentiated and differentiated states are not sufficient for the quantitative characterization of the continuous differentiation processes

Using the simplest readouts to quantify and delineate different biological conditions/states is always preferred because it provides

more direct insight regarding the underlying mechanisms. Is it possible that our approach is overly complicated and exceeds what is required to quantitatively describe the differentiation process? Are straight-forward single-cell measurements sufficient to discriminate between undifferentiated and differentiated cells and follow the differentiation process? To test this possibility, we evaluated the discriminative performance of single-cell properties that are expected to deviate between the undifferentiated and differentiated cells. These included cell speed, actin intensity, the temporal derivative of actin intensity, migration persistence, and local density. The local density dramatically increased over time for cells grown in proliferation medium due to continued proliferation throughout the experiment (Fig. 3A). The mean speed and actin intensity in proliferating cultures slightly decreased and increased correspondingly over time, perhaps due to the increased density (Fig. 3B,C), and the mean temporal derivative of actin intensity fluctuated around zero for both differentiating and proliferating cells (Fig. 3D). Persistent migration of proliferating cells was lower compared to differentiating cells without a clear trend over time (Fig. 3E). Each of these four discriminative readouts, as well as their integration, could be generalized across experiments as demonstrated by using each feature to train a machine-learning model and applying this model to discriminate between the two experimental conditions in an independent experiment (Fig. 3F).

The model trained with the local density and the model trained with all four features surpassed the discrimination performance of the time-series motility and actin models (also reported in Fig. 2B,C). However, discrimination does not necessarily imply that these readouts can be used to quantitatively describe the differentiation process. Indeed, the differentiation score of each of these classifiers could not capture the differentiation process as measured by single-cell monotonic increase at the critical differentiation time interval of 7.5–14.5 h. The single-cell correlations between the differentiation score and time were low for all the single-feature classifiers, as well as for the integrated classifier (Fig. 3G). This is in contrast to our classifiers that generalized to effectively quantify the differentiation process leading to a higher correlation between the differentiation score and time (Fig. 3G—motility, actin intensity, same data as in Fig. 2G). A plausible explanation for why these effective discriminating models could not capture the continuous differentiation process is that the discriminating features captured properties attributed to the undifferentiated state. For example, the increased local cell density of

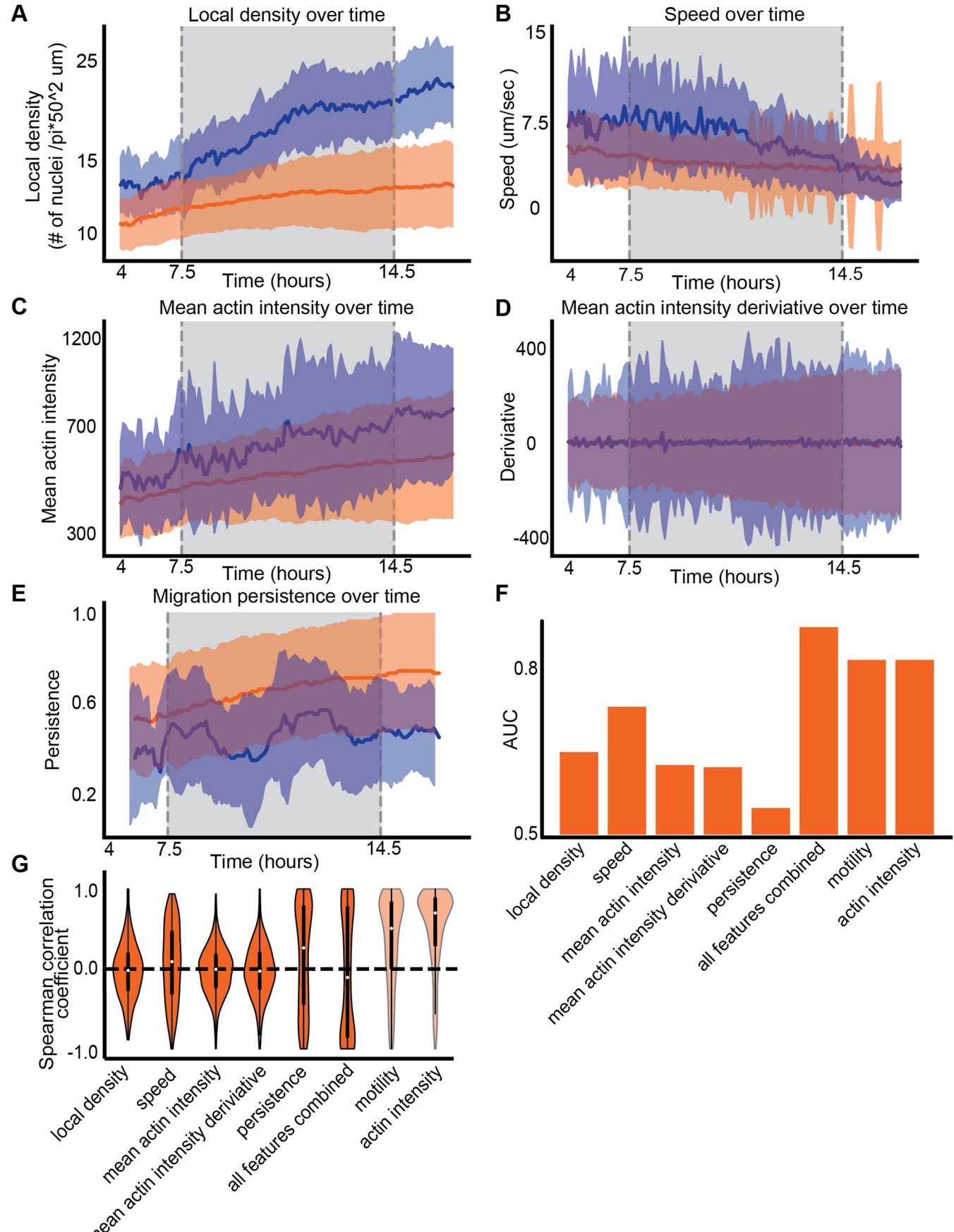

◄ **Figure 3. Simple single-cell measurements are insufficient for continuous cell state transition characterization.**

(A–E) Mean (solid line) and standard deviation (shade) of single-cell characteristics over time of ERKi-(orange) and DMSO- (blue) treated cells. $N = 575$ cells. Single-cell properties included are local cell density (**A**), mean speed (**B**), mean actin intensity (**C**), mean actin intensity temporal derivative (**D**), and persistence in migration (**E**). (**F**) Single-cell-state classification performance. Area under the receiver operating characteristic (ROC) curve (AUC) for classifiers trained using local density, speed, mean actin intensity, mean actin intensity derivative, persistent migration, and integration of these properties to discriminate ERKi- and DMSO-treated cells experimental conditions. The right-most bars show AUCs for classifiers trained with motility and actin dynamics. AUC scores were calculated for 757 temporal segments of differentiated/undifferentiated cells from an experiment that was not used for model training (full details in "Methods"). (**G**) The single-cell correlation distribution between the differentiation score and time for all the classifiers is shown in (**F**) (median shown in white). Dashed horizontal line shows no correlation. The thick black bar represents the interquartile range between Q1 and Q3, the thin black line extending from it represents the rest of the distribution where its edges show the maximal and minimal values, and the white dot shows the median correlation. The right-most distributions show correlations for classifiers trained with motility and actin dynamics ($N = 575$ cells).

proliferating cells can be used to effectively discriminate between the two populations but does not provide any information regarding the progression through differentiation. Indeed, training models that included temporal features extracted from single-cell local density dynamics showed the same or deteriorated correlation between the differentiation score and time compared to models that were not trained with local density (Appendix Fig. S12). Motility persistence, which was the most monotonic among the single feature, and was earlier identified to be associated with differentiation (Fig. EV4), was still far behind the integrated models in terms of monotonicity, suggesting that there is further discriminative information beyond persistence in the cells' trajectories. These results indicate that effective discrimination between the discrete extreme states is insufficient for the quantitative characterization of continuous state transitions. Specifically, using machine learning for quantitative characterization requires extracting features that can capture the state transition and avoiding features that may confound the quantitative characterization of the process (e.g., avoiding local cell density in characterizing the differentiation process). In our case, and in agreement with other studies (Copperman et al, 2021; Wang et al, 2020; Wu et al, 2022), integration of multiple dynamic features encoding the temporal changes were necessary to continuously measure a biological process.

## The transition from terminal differentiation to fusion is controlled by p38

We next aimed at harnessing our single myoblasts continuous differentiation scores to investigate the relationship between cell differentiation and fusion. We manually annotated the fusion time of 68 myoblasts that fused to 6 myofibers (Fig. 4A) and used the continuous differentiation score to determine an estimated time of the terminal differentiation state. Both the distributions of the single cells' terminal differentiation and fusion times followed a normal-like distribution, where the variability in the predicted differentiation time was higher than that of fusion time (Fig. 4B). The time duration between terminal differentiation and fusion also followed a normal-like distribution, indicating a typical duration of ~3 h between terminal differentiation and fusion at the population scale (Fig. 4C). The mean of a ~3 h gap between predicted terminal differentiation and fusion is not trivially derived from the definition of differentiation timing during training because (1) the differentiation time at training was defined as 2.5 h before the first fusion event, and (2) the heterogeneity in fusion timing spans over ~10 h (Fig. 4B). These results suggest that cells undergo fusion within a typical time interval from their terminal differentiation. This

coupling was validated by measuring a correlation between single-cell differentiation and fusion times (Fig. 4D) and was not sensitive to the threshold used to determine the terminal differentiation time (Appendix Fig. S13). These results suggest that myoblasts must reach a differentiation checkpoint before fusion can proceed.

Previous studies have shown that co-inhibition of p38, a family of MAP kinases that play a critical role in the initiation of the differentiation program, together with a promyogenic factor, overcomes the early block in differentiation but not the later impairment of muscle cell fusion imposed by the p38 inhibitor, leading to differentiated unfused cells (Gardner et al, 2015). Following this logic, we treated primary myoblasts with the promyogenic ERKi and the p38 inhibitor BIRB 796 (p38i; 5 μM) and performed primary myoblasts live imaging experiments. There was little appreciable difference between cells treated with p38i and controls treated with DMSO, consistent with previous studies showing that p38i maintains myoblasts in a proliferative undifferentiated state (Zetser et al, 1999). Myoblasts co-treated with ERKi and p38i appeared differentiated but failed to fuse, leading to the complete absence of multinucleated myofibers, reinforcing the notion that p38 is essential for a transition from differentiation to fusion and maturation (Fig. 5A,B). Immunofluorescence staining validated that the fraction of MyoG-positive cells remained low for p38i-treated cells and increased in cultures co-treated with p38i and ERKi, indicating that co-inhibition of p38 and ERK1/2 leads to bona fide differentiation (Figs. 5C and EV2; Movie EV3). Moreover, cells started fusing once the p38 inhibitor was washed out, indicating that they were stalled at a "ready to fuse" state (Fig. EV5).

However, it was not clear whether the differentiation process was altered with respect to ERKi-treated cells. Thus, we quantitatively described the differentiation process of cells co-treated with p38i and ERKi by applying our motility and actin models trained with proliferating and differentiating cells. As a control, we validated that the differentiation score profile of p38i-treated cells resembled that of proliferating cells treated with DMSO alone (Fig. 5D,E; Appendix Figs. S14 and S15). The motility classifier showed that the differentiation profile of p38i-ERKi-treated cells followed a trend strikingly similar to the one obtained for ERKi-treated cells and specifically included the gradual transition at the critical time of 7.5–14.5 h (Fig. 5D). The mean actin classifier score maxed at a value of ~0.5, which was not sufficient to distinguish undifferentiated from differentiated cells, suggesting that p38i alters actin dynamics (Fig. 5E). Nevertheless, the actin model predicted a monotonically increasing trend, suggesting that the cells were becoming more differentiated over time.

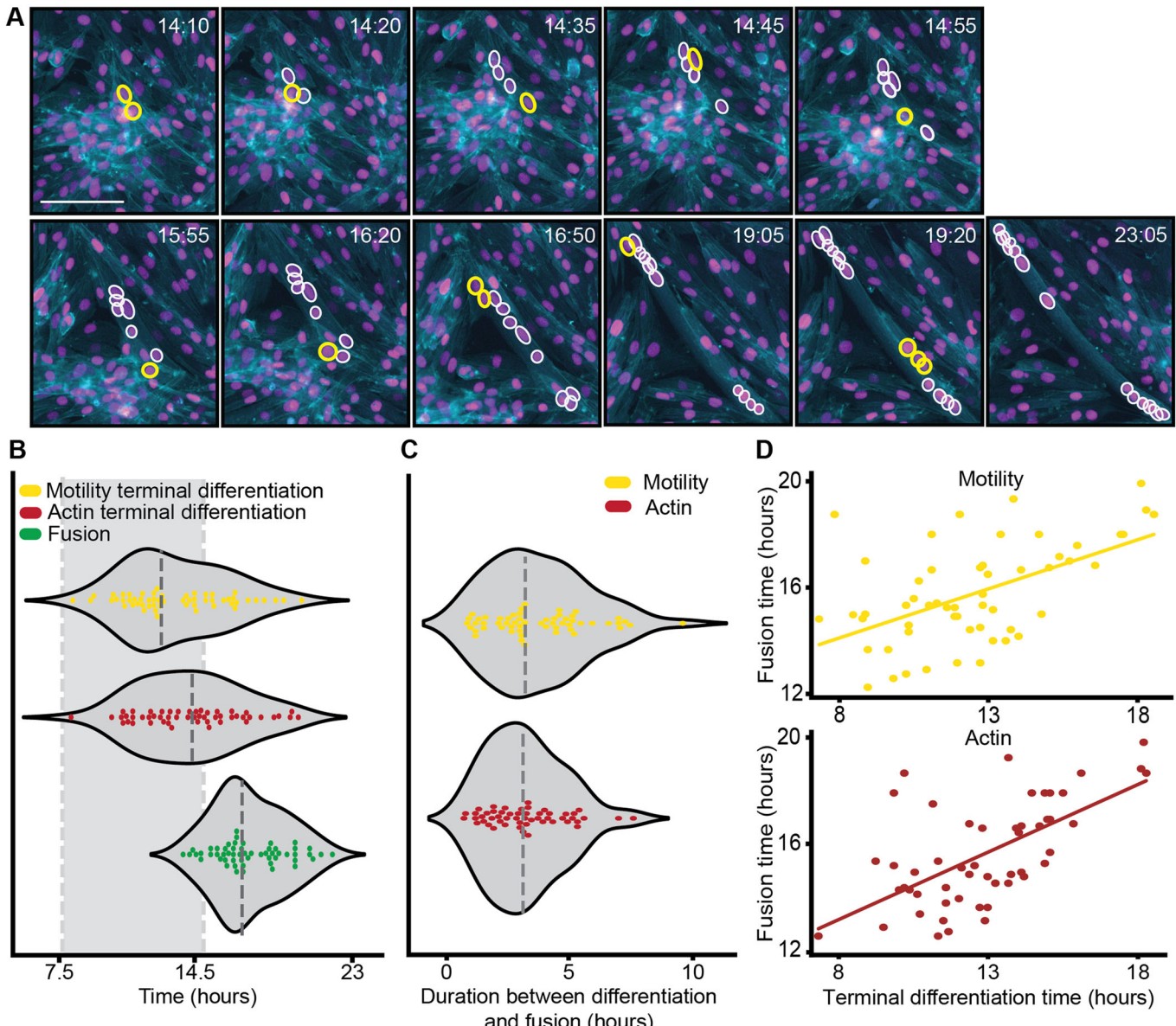

**Figure 4. Correlation between terminal differentiation and fusion time.**

(A) Annotations of single-cell fusion into a representative myofiber over time. Cells marked in white are already fused, and cells marked in yellow are fusing into the fiber. Scale bar 100 μm. (B) Distribution of single cells' fusion times (green) and terminal differentiation times determined by motility (yellow) and actin intensity (red) classifiers. The dashed vertical gray rectangle highlights the differentiation time interval of 7.5–14.5 h. All three distributions were normal-like as assessed by the D'Agostino's K-squared test not rejecting the null hypothesis that the terminal differentiation time was normally distributed (D'Agostino's K-squared test: motility classifier: P value = 0.36, actin classifier P value = 0.64; fusion time P value = 0.1). The "terminal differentiation" state was determined using a differentiation score threshold of 0.78 (the same threshold was also used in panels C-D). The models identified 56 (motility) and 52 (actin intensity) cells that reached a terminal differentiation state, out of 68 annotated cells. 71% (motility) and 65% (actin intensity) of the identified cells reached a terminally differentiated state by 15 h post induction. The median time of terminal differentiation was 12.63 (motility) and 14.2 (actin intensity); the median fusion time was 16.8 h. (C) Distribution of the duration between single-cell terminal differentiation and fusion, for terminal differentiation determined by motility (yellow) and actin (red) classifiers (N = 68 cells). Both distributions were normal-like as assessed by the D'Agostino's K-squared test not rejecting the null hypothesis that the duration was normally distributed (D'Agostino's K-squared test: motility classifier: P value = 0.13, actin classifier: P value = 0.13). Median differentiation-to-fusion duration was 3.1 (motility) and 3 (intensity) hours. (D) Associating single-cell terminal differentiation time (x axis) and fusion time (y axis), determined by the motility (yellow) and the actin (red) classifiers. Pearson correlation coefficients were 0.52 (motility) and 0.73 (actin intensity), Pearson correlation's P value < 0.001 for both actin and motility classifiers (N = 68 cells).

To test the model's prediction that co-inhibition of p38 and ERK1/2 leads to properly differentiated cells that are ready to fuse and that actin dynamics are altered under these conditions, we acquired mass spectrometry data. We extracted proteins from primary myoblasts treated with ERKi, ERKi+p38i, p38i, and DMSO control for 24 h and used mass spectrometry-based proteomics for unbiased, high-throughput identification of differentially expressed proteins. In total 4319 proteins were identified

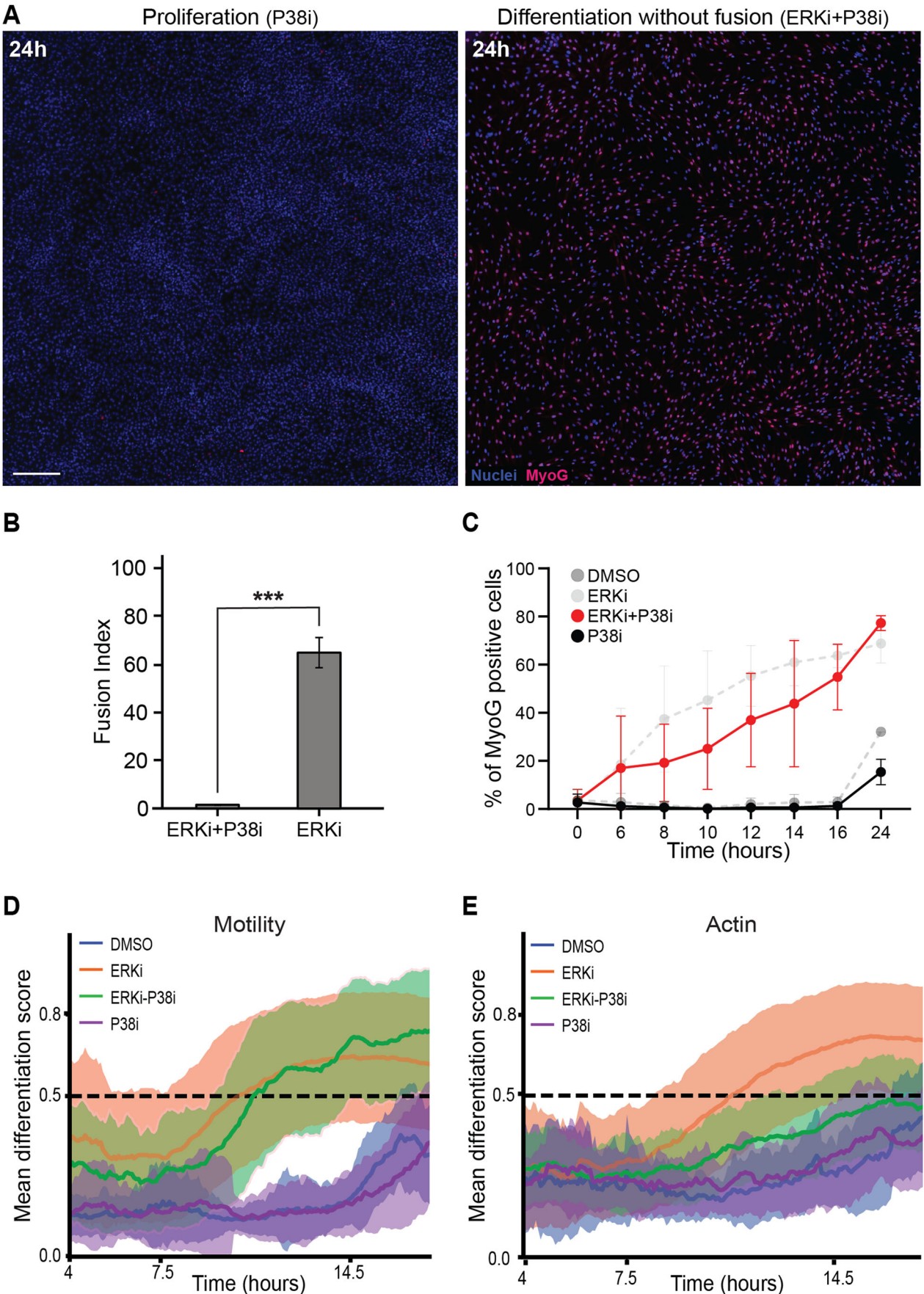

**Figure 5. Differentiation and fusion can be uncoupled.**

(A) Overlay images of primary myoblasts treated with p38i (left) and ERKi+p38i (left), fixed at 24 h and stained for MyoG (red) and nuclei (Hoechst, blue). Magnification ×5. Scale bar: 100 μm. Overlay images of myoblasts undergo differentiation but do not fuse. Scale bar 100 μm. (B) Fusion Index: Percentage of fused nuclei in ERKi and ERKi-p38i-treated cultures. Error bar represents standard deviation (SD) of the mean ($n > 50,000$ cells per condition). Two-sample Student's $t$ test was used to assess statistical differences between the groups, *** indicate statistical significance $P$ value < 0.0005. (C) Percentage of MyoG-positive cells over time under proliferation conditions (p38i; Black) and differentiation conditions (ERKi + p38i; red). DMDO and ERKi from Fig. 1B are shown for convenience (semitransparent). The error bar represents the standard deviation (SD) of the mean ($n > 50,000$ cells per condition). (D, E) Mean (solid line) and standard deviation (shade) of the differentiation score over time of ERKi- (orange), DMSO- (blue), ERKi+p38i- (green), and p38i- alone (purple)-treated cells using the motility (C) and actin intensity (D) classifiers (ERKi: 575 cells; ERKi+p38i: 208 cells; p38i: 202 cells; DMSO: 103 cells). ERKi- and DMSO-treated cells differentiation scores are the same as in Fig. 2. The analysis for the entire experiment is shown at (Appendix Figs. S14 and S15).

across all experiments. Unsupervised hierarchical clustering analysis showed that p38i-treated cultures cluster with the DMSO proliferation control, consistent with previous studies showing that p38 inhibition maintains myoblasts in the proliferative undifferentiated state (Fig. 6A). The analysis also showed that ERK1/2 and p38 co-inhibited cultures cluster with the differentiating ERKi-treated cultures, suggesting that differentiation occurs under co-inhibition conditions (Fig. 6A).

The actin cytoskeleton is essential for fusion and for the subsequent formation of sarcomeres (Abmayr and Pavlath, 2012; Deng et al, 2015; Gruenbaum-Cohen et al, 2012; Onel and Renkawitz-Pohl, 2009; Richardson et al, 2008). Gene Ontology (GO) annotation enrichment analysis showed a significant upregulation of proteins involved in the process of actin cytoskeleton organization and downregulation in the process of skeletal muscle contraction and maturation in the ERKi-p38i co-inhibition cultures compared to ERKi alone, consistent with an arrest prior to fusion and maturation (Fig. 6B). These results supported our conclusion that co-inhibited cells arrest at terminal differentiation before fusion and maturation and explain the deviation in the differentiation score of the actin, but not the motility classifier, upon ERKi-p38i co-inhibition.

Strikingly, only 18 out of 504 proteins that were significantly differentially expressed between the differentiated (ERKi) and the differentiated unfused cells (ERKi-p38i) were completely missing from the differentiated unfused cells (Fig. 6C). One of these being Myomixer, which is known to be essential for fusion (Zhang et al, 2017; Quinn et al, 2017; Bi et al, 2017). These results suggest that cell differentiation and fusion can be uncoupled and that p38 is essential for the transition from terminal differentiation to fusion and maturation, supporting the notion that differentiation and fusion are coordinated processes that occur in tandem.

## Discussion

We combined live-cell imaging and machine learning to infer the differentiation state of single cells during the process of muscle precursor cell differentiation. Many studies highlight the rich information encapsulated in single-cell dynamics that, with the aid of supervised or unsupervised machine learning, enable effective identification of sub-populations and discrimination of perturbations (Choi et al, 2021; Goglia et al, 2020; Jacques et al, 2021; Jena et al, 2022; Kimmel et al, 2018; Valls and Esposito, 2022), that cannot be inferred from static snapshot images (Copperman et al, 2021; Wang et al, 2020; Wu et al, 2022). For example, approaches that rely on static snapshots make it extremely hard to infer

trajectories that deviate from the mainstream cell-state progression because they are confounded by cell-to-cell variability. The ability to measure a single-cell state as it transitions through time during a physiological process, along with careful experimental–computational interplay, enabled us to quantitatively follow the process and derive biological insight. Specifically, identify the key time frame where myoblasts gradually undergo differentiation (Fig. 2D,E), link single-cell differentiation to fusion (Fig. 4D), associate persistent migration with differentiation (Fig. EV4), validate that co-inhibition of p38 and ERK1/2 arrests the fusion process without altering the differentiation process, and confirm our model's prediction that actin regulators are altered (Figs. 5 and 6). Ultimately, our experiments show that p38 regulates the transition from differentiation to fusion, implying that there is a differentiation checkpoint that must be reached before fusion occurs, and opening new avenues to identify novel regulators of this process.

The ability to infer the differentiation state of individual myoblasts can further enable the identification of novel myogenic factors, high-throughput screening for pro-regenerative compounds, and the definition and subsequent examination of distinct intermediate steps in the differentiation process. Moreover, this approach of harnessing temporal dynamics by machine learning, without explicit state markers, can be generalized beyond terminal differentiation. Such a computational estimation of the cell state may have wide applications in characterizing other single-cell dynamic functions such as transitioning during the cell cycle, epithelial to mesenchymal transition, immotile to motile, disease progression, and cell death. The dynamic state readout can be correlated to other, independently measured cell readouts to systematically characterize the full spectrum of heterogeneities in complex biological processes.

Unsupervised approaches for cell-state inference traverse from an initial to a final state through steps that rely on similarity in cell appearance (Gut et al, 2015). These trajectories can be distorted by batch effects or cell phenotypes unrelated to the state transition. In our approach, the supervised component forces the trajectory to follow the phenotypic axes most relevant to the state transition under investigation. This approach is similar to the approaches taken by (Szkalisity et al, 2021), which rely on the manual assignment of cells to discrete states in 2D that are then inferred by regression analysis, or by (Stallaert et al, 2022) that uses a supervised model to select features predictive of the cell state before constructing cell-state trajectories.

Our approach uses the physiological cell state (undifferentiated vs. differentiated) as the ground truth, optimizes binary classification, and uses the classification's confidence score as the cell-state

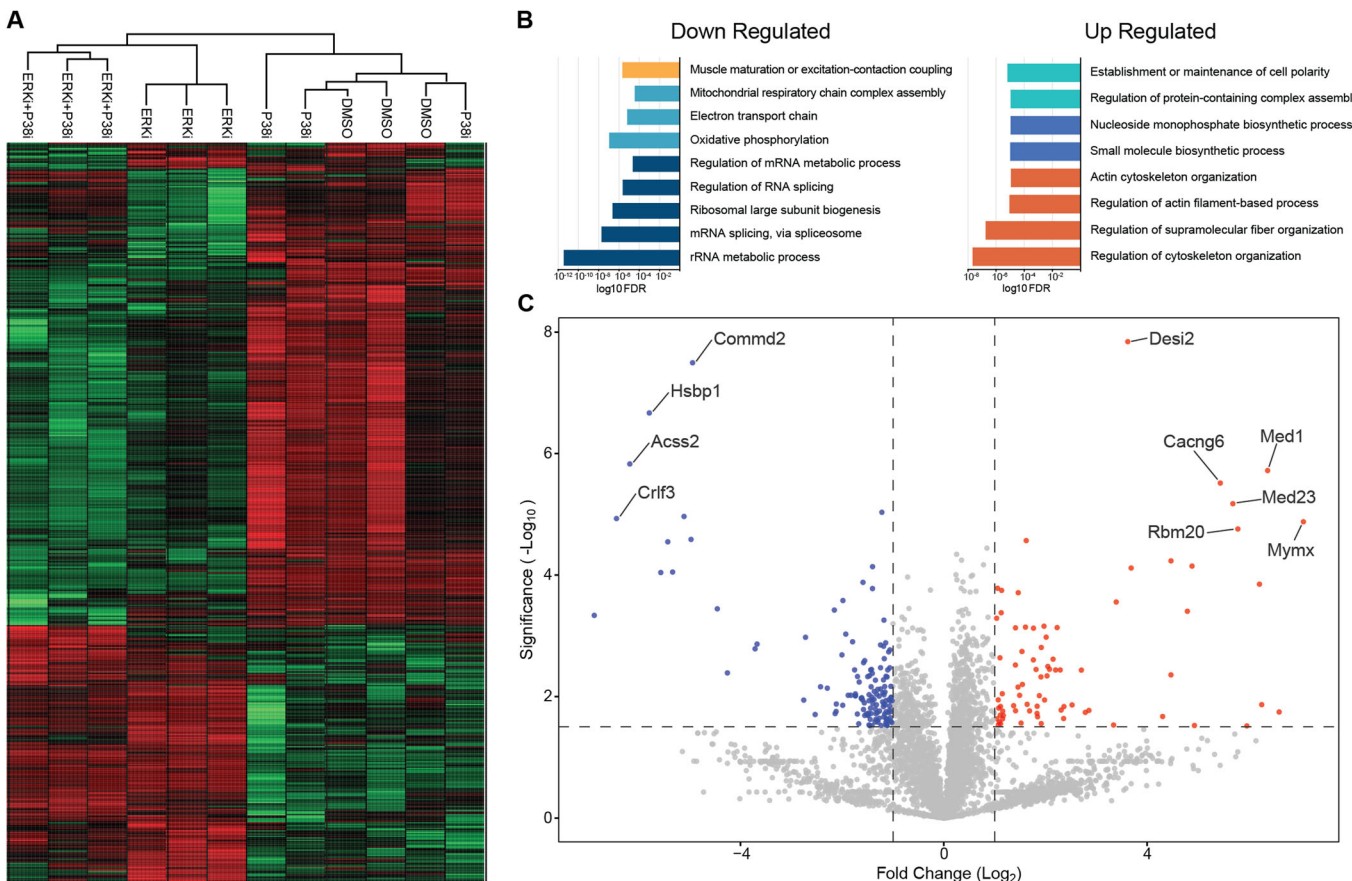

**Figure 6. Mass spectroscopy analysis reveals differential protein expression between ERK1/2 inhibition and p38-ERK1/2 co-inhibition.**

(A) Heatmap of significantly changed proteins (fold change > 2 and *P* value < 0.05) between ERK1/2 inhibition samples and p38-ERK1/2 co-inhibition samples, after unsupervised hierarchical clustering. Each column represents the sample and each row represents the differentially expressed protein. High expression marked in red; low expression marked in green. The sample clustering tree is shown on the top. (B) Bar plot of Gene Ontology (GO) Annotation Enrichment Analysis comparing ERK1/2 inhibition samples and p38-ERK1/2 co-inhibition samples. Colors denote different protein groups. (C) Volcano plot of differentially expressed proteins between ERK1/2 inhibition samples and p38-ERK1/2 co-inhibition samples. In total, 4317 proteins are shown, upregulated (red) and downregulated (blue). $N = 3$ biological replicates. One-way ANOVA was used to assess statistical significance among the four experimental groups, the statistical analysis was done using Perseus software.

measurement. However, there is no guarantee that the classification's confidence score has linear properties. For example, whether the difference in scores between 0.3 and 0.4 has the same phenotypic magnitude as between scores of 0.6 and 0.7. This limitation is also common to approaches that use non-linear dimensionality reduction (Copperman et al, 2021; Eulenberg et al, 2017; Jacques et al, 2021; Rappez et al, 2020; Stallaert et al, 2022; Wang et al, 2022; Wu et al, 2022) and could also limit unsupervised state representations that can be dominated by features that do not relate to the cell state (Copperman et al, 2021; Jacques et al, 2021; Wang et al, 2022; Wu et al, 2022). Still, the monotonicity property holds, e.g., a differentiation score of 0.4 is predicted to be more advanced along the differentiation trajectory than a differentiation score of 0.3. This implies that the machine-learning model captures more phenotypic evidence for the advancement along the state transition axis. This monotonicity property is sufficient for comparing different trajectories and calculating temporal correlations between cell state and other properties, as demonstrated here, with motility persistence, and elsewhere (e.g., Mayr et al, 2021; Zaritsky et al, 2021).

## Methods

### Mouse lines

We used 6–8 weeks-old female Actin and nuclear reporter mice (LifeAct-GFP/ nTnG[+/+]) (Eigler et al, 2021). Fluorescence expression was validated using visual inspection. All experiments were approved by the Animal Care and Use Committee of the Weizmann Institute of Science (IACUC application #07690920-3).

### Isolation and treatment of primary myoblasts

Primary mouse myoblasts were isolated from gastrocnemius muscle using mechanical tissue dissociation as in ref. Eigler et al, 2021. Briefly, after cutting the muscle tissue into small pieces, they were incubated in Trypsin EDTA Solution B (0.25% Trypsin and 0.5% EDTA, Biological Industries Israel) and subjected to mechanical dissociation with a serological pipet. Supernatants were strained (FALCON REF no. 352340) and centrifuged. Cell pellets were resuspended in BioAMF™-2 media (Biological Industries, Israel),

plated on 10% Matrigel® Matrix (Corning REF no. 354234) coated plates, and grown at 37° in a 5% $CO_2$ incubator. BioAMF™-2 was used in all experiments (Biological Industries Israel).

## Microscopy

For live imaging, 40.000 cells were plated in a Slide eight-well chamber (ibidi GmbH, cat. no. 80826) coated with 10% Matrigel® Matrix. Fifteen hours after cell seeding, the different treatments were added to the cells cultured in proliferation medium BioAMF™-2 (Biological Industries Israel). To induce myoblasts differentiation, cells were treated with 1 μM ERK inhibitor (SCH 772984 Cayman Chemical Company). The inhibitors are dissolved in Dimethyl Sulfoxide (DMSO, MP Biomedicals cat. no. 196055, 1.10 g/ml stock concentration). Therefore, in the control sample of proliferation, DMSO treatment was added in a concentration of 1 μg/ml (equal to 1 μl, the volume added of each inhibitor). In the samples treated with p38 inhibitor, were used 5 μM (BIRB 796, AXON 1358) either alone or together with ERKi.

Live imaging (37 °C, with 5% $CO_2$) was performed using Cell discoverer 7-Zaiss inverted in widefield mode with Zeiss Axiocam 506 camera Carl Zeiss Ltd. Images were acquired using a ZEISS Plan-APOCHROMAT 20×/0.70 Autocorr Objective (Working distance 2.20 mm). Excitation 470 nm for GFP signals (LifeAct) and 567 nm for tdTomato (nuclei). ZEN blue software 3.1 was used for image acquisition. If necessary, linear adjustments to brightness and contrast were applied using ImageJ v1.52 software (Schindelin et al, 2012). Cells were imaged 1.5 h after adding the treatments, with 5 min intervals and at a pixel size of 0.462 μm.

Fixed samples were imaged using a ZEISS Plan-APOCHROMAT 5×/0.35 Autocorr Objective (Working distance 5.10 mm), 1.178 μm/px. Excitation 470 nm for GFP (MyHC) and 567 nm for Alexa Fluor® 568 (MyoG) and 405 nm for nuclei stained with Hoechst 33342.

## Immunofluorescence staining of MyoG-MyHC

Primary myoblasts were seeded in a 96-well culture dish, coated in Matrigel® Matrix at 8000 cells per well cultured in BioAMF-2 media. After 15-h incubation at 37 °C in a 5% $CO_2$ incubator, the cells were treated with 1 μM ERK inhibitor (SCH 772984 Cayman Chemical Company) and 5 μM p38 inhibitor (BIRB 796, AXON 1358) in the needed samples. The inhibitors are dissolved in Dimethyl Sulfoxide (DMSO, MP Biomedicals cat. no. 196055, 1.10 g/ml stock concentration). Therefore, in the control sample of proliferation, DMSO treatment was added in a concentration of 1 μg/ml (equal to 1 μl, the volume added of each inhibitor). Cells were fixed at specific time points (0 h–6 h–8 h–10 h–12 h–16 h–24 h) with 3.7% PFA in PBS for 15 min at room temperature. The cells were then quenched with 40mM ammonium chloride for 5 min, washed with PBS three times, permeabilized in PBS with 0.01% Triton x-100 for 10 min, and blocked in 10% FBS in PBS (blocking buffer) for 1 h at room temperature. Primary antibody incubation was done in a blocking buffer overnight at 4°, with the following antibodies: Anti-Fast Myosin Heavy Chain antibody [MY-32] (Ab51263), Abcam) 1:400, Anti-Myogenin antibody [EPR4789] (ab124800) 1:500. Cells were washed three times in PBS and then incubated with secondary antibodies: Goat Anti-Mouse IgG H&L (Alexa Fluor® 488) (ab150117) 1:600, Donkey Anti-Rabbit IgG H&L (Alexa Fluor® 647) (ab150067) 1:600, Donkey Anti-Rabbit

IgG H&L (Alexa Fluor® 568) (ab175692) 1:600. The cells were washed three times in PBS, incubated with Hoechst 33342 (Thermo Scientific cat. no. 62249, 1:1000) for 5 min and washed in PBS.

### Quantification

The percentage of expressing cells was calculated by dividing the number of nuclei labeled by the MyoG antibody by the total amount of cells given by the Hoechst staining in three independent replicates of each experimental condition. The nuclei were segmented and counted using the Cellpose software (Stringer et al, 2021).

### Quantification of fusion index

First, the nuclei were segmented and counted using the Cellpose software (Stringer et al, 2021) together with a homemade Python script to gain the total number of nuclei. Then, the fusion index was quantified by manually identifying the number of nuclei found in cells with at least two nuclei. The values were expressed as a percentage of the total number of nuclei per field of view.

### Actin intensity quantification in a field of view

The quantification was made using the ImageJ v1.52 software (Schindelin et al, 2012). We measured the fluorescence intensity signal of the entire field of view every hour and plotted the mean intensity with stdDev calculated over all the pixel values of every field of view.

## Protein sample preparation for mass spectrometry

Primary myoblasts were grown on 10-cm plates coated in Matrigel® Matrix at $1.2 \times 10^6$ cells per plate, cultured in BioAMF-2 media. After 15 h the cells were treated with: (1) 1 μM ERK inhibitor (SCH 772984 Cayman Chemical Company) (2) 5 μM p38 inhibitor (BIRB 796, AXON 1358) (3) 1 μM ERK inhibitor with 5 μM p38 inhibitor and (4) 0.1% DMSO in the control samples. Twenty-four hours after treatment, cells were washed with PBS, harvested by scraping and pelleted by centrifugation at 2000×*g* for 5 min. We lysed each pellet in 1% SDS lysis buffer: 1% SDS in 50 mM Tris (pH 8) containing protease inhibitor cocktail (Sigma Aldrich, catalog no. P8849) and 1 mM PMSF. The samples were boiled at 95 °C for 5 min, then diluted into 2× RIPA lysis buffer to give 1× RIPA lysis buffer (50 mM Tris-HCl [pH 8.0], 150 mM NaCl, 0.5% SDS, 0.5% sodium deoxycholate, 1% Triton X-100). The lysates were cleared by centrifugation at 16,000×*g* for 10 min at 4 °C. Protein concentration analysis was done using BCA kit (Pierce). A total of 12 individual cell pellets were prepared for LC MS\MS analysis— four treatments with three biological repeats. Protein samples were frozen in liquid nitrogen and stored at −80 °C.

## Proteolysis and mass spectrometry analysis

The proteins were dissolved in RIPA buffer, sonicated and precipitated in 80% acetone. The protein pellets were dissolved in in 9M Urea and 400 mM ammonium bicarbonate than reduced with 3 mM DTT (60 °C for 30 min), modified with 10 mM iodoacetamide in 100 mM ammonium bicarbonate (room temperature 30 min in the dark) and digested in 2M Urea, 25mM ammonium bicarbonate with modified trypsin (Promega), overnight at 37 °C in a 1:50 (M/M) enzyme-to-substrate ratio. The

tryptic peptides were desalted using C18 (Top tip, Glygen) tip, dried and resuspended in 0.1% formic acid.

The peptides were resolved by reverse-phase chromatography on 0.075 × 300-mm fused silica capillaries (J&W) packed with Reprosil reversed-phase material (Dr Maisch GmbH, Germany).

The peptides were eluted with linear 180 min gradient of 5 to 28% acetonitrile with 0.1% formic acid in water, 15 min gradient of 28 to 95% and 25 min at 95% acetonitrile with 0.1% formic acid in water, at flow rates of 0.15 µl/min. Mass spectrometry was performed by Q Exactive HFX mass spectrometer (Thermo) in a positive mode ($m/z$ 300–1800, resolution 120,000 for MS1 and 15,000 for MS2) using repetitively full MS scan followed by high collision induces dissociation (HCD, at 27 normalized collision energy) of the 30 most dominant ions (>1 charges) selected from the first MS scan. The AGC settings were $3 \times 10^6$ for the full MS and $1 \times 10^5$ for the MS/MS scans. The intensity threshold for triggering MS/MS analysis was $1 \times 10^4$. A dynamic exclusion list was enabled with an exclusion duration of 20 s.

The mass spectrometry data was analyzed using the MaxQuant software 1.5.2.8 (1) for peak picking and identification using the Andromeda search engine, searching against the mouse proteome from the Uniprot database with mass tolerance of 6 ppm for the precursor masses and 20 ppm for the fragment ions. Oxidation on methionine and protein N-terminus acetylation were accepted as variable modifications and carbamidomethyl on cysteine was accepted as static modifications. Minimal peptide length was set to six amino acids and a maximum of two mis cleavages was allowed. Peptide- and protein-level false discovery rates (FDRs) were filtered to 1% using the target-decoy strategy. The data were quantified by label-free analysis using the same software with "match between runs" option. Protein tables were filtered to eliminate the identifications from the reverse database, and common contaminants. Statistical analysis of the identification and quantization results was done using Perseus 1.6.10.43 software. Annotation enrichment was done by the string tool (https://string-db.org/) and the David bioinformatics package (https://david.ncifcrf.gov/).

## Automated single-cell tracking and quantification

Automatic nuclei speed was performed using the commercial software Imaris (v9.7.2, Oxford Instruments). We created a new "spots" layer on the nuclei label channel using the default Favorite Creation Parameters to track the spots over time, classify the spots, and object-object statistics. Next, we estimated the diameter of 8 µm and enabled background subtraction. These analyses allowed us to collect a large number of single-nuclei trajectories. While trajectories frequently fragment using this approach, they were sufficient to quantify the mean nuclei speed over time.

## Semi-manual single-cell tracking

Semi-manual single-cell tracking was performed to obtain accurate trajectories for training and evaluating our machine-learning models. The time-lapse images were first converted to XML/hdf5 format using the BigDataViewer (v.6.2.1) FIJI plugin (Pietzsch et al, 2015; Schindelin et al, 2012). We then used the Mastodon FIJI plugin (Mastodon—a large-scale tracking and track-editing framework for large, multi-view images; https://github.com/mastodon-sc/mastodon), for single-cell tracking and manual correction. We

tracked cells that resided within the field of view throughout the entire experiment and included cells that fused into multinucleated fibers and cells that did not fuse within the experimental time frame. To reduce the manual annotation load, tracks that contained less tracking errors were prioritized for manual correction. Altogether, we collected 848 tracks for training (538 ERKi-treated cells; 310 DMSO-treated cells), 789 tracks, from an independent experiment, for testing (686 ERKi-treated cells; 103 DMSO-treated cells), and 410 tracks, from the perturbation experiment (202 p38i-treated cells; 208 ERKi+p38i-treated cells).

## Preprocessing trajectories

We used OpenCV's CalcOpticalFlowFarneback, based on Gunner Farneback's method (Farnebäck, 2003), for image registration to correct erroneous offsets of the tracked cells' trajectories. For each pair of frames, we calculated the average offset and used the corresponding translation for registration.

## Models training

The training pipeline implements the following steps.

1. Determining labels for training. We assigned ERKi-treated cells with the "differentiated" label in a time segment of 2.5 h (hours 12.3–14.8) before the first fusion event was observed in the field of view. We decided not to label ERKi-treated cells as "undifferentiated" at the onset of the experiment because we did not know how early differentiation phenotypic signs appear. The increase in MyoG-positive cells during the first 6 h of the experiment supports this decision. We assigned time segments of DMSO-treated cells with the "undifferentiated" label because their differentiation begins after more than 23 h of the experiment.
2. Partitioning single-cell trajectories to temporal segments. We partitioned trajectories of DMSO- and ERKi-treated cells to overlapping temporal segments (overlap lag = 5 min) in equal lengths of 2.5 h each. Temporal segments' length was determined to match the time frame where we consider ERKi-treated cells as "differentiated". This step resulted in 16,636 DMSO-treated cells and 47,819 ERKi-treated cells temporal segments. For training, we labeled as "differentiated" ERKi-treated cells in the temporal segment of hours 12.3–14.8 and labeled as "undifferentiated" DMSO-treated cells in non-overlapping temporal segments throughout the experiment. Overall, we extracted 468 undifferentiated and 268 differentiated temporal segments for training.
3. Extracting motility and actin features. We extracted single-cell motility and actin intensity time series from each temporal segment:
   - Motility: We calculated the displacement of a single cell for each time point $t$, creating a two-dimensional vector: $displacement(t) = (x_t - x_{t-1}, y_t - y_{t-1})$
   - Actin: We cropped a quantification window of size $32 \times 32$ µm around the center of each nucleus at each time point and calculated the minimum, maximum, mean, median, and standard deviation of the actin intensity within the window.
4. Extracting hundreds of single-cell time-series features using the "tsfresh" python package (Christ et al, 2018). These features encoded properties of the temporal segments, such as temporal peaks, derivatives, and statistics. The tsfresh feature selection was

based on the Benjamini–Yekutieli multiple test procedure (Benjamini and Yekutieli, 2001) to identify the most relevant features for characterizing the time series.

5. Training classifiers to distinguish between differentiated and undifferentiated cells. We trained random forest classifiers, which are considered effective with high dimensional and relatively small datasets (Breiman, 2001), as validated empirically on our data (Fig. S4). Hyperparameter tuning was performed using a grid search with fivefold cross-validation (motility classifier: {"max_-depth': 12, 'min_samples_leaf': 1, 'n_estimators': 100}, actin intensity classifier: {'max_depth': 20, 'min_samples_leaf': 1, 'n_estimators': 200}).

6. Evaluating the trained classifies' performance. We assessed the discrimination performance of our motility/actin classifiers on an independent experiment that was not used for training. We partitioned time series to overlapping temporal segments (102,929 ERKi-treated cells segments, 7214 DMSO-treated cells segments), selected temporal segments for evaluation as described above for 577 differentiated and 180 undifferentiated temporal segments, extracted motility and actin intensity time series, performed feature extraction using "tsfresh", and evaluated the performance of the corresponding trained models. The AUC of the motility and the actin intensity classifiers were 0.8 and 0.81, correspondingly (Fig. 3F); accuracy was 0.76 and 0.81; precision was 0.97 and 0.93; recall was 0.71 and 0.81.

## Inference of single cells differentiation trajectories

Each single-cell trajectory was partitioned into overlapping temporal segments of 2.5 h, with an overlapping lag of 5 min (one frame). We calculated motility & actin intensity time series, applied "tsfresh", selected features according to training, and applied the corresponding trained models on these feature vectors to retrieve a differentiation score for each segment defining single-cell differentiation trajectories.

## Random field theory (RFT)-based inference

To evaluate the statistical significance of differentiation score differences among cells subjected to distinct experimental conditions, we employed random field theory (RFT) (Pataky et al, 2015; Pataky 2016). RFT requires equivalent trajectory lengths. Thus, the variation in the temporal spans of the tracked single-cell trajectories led us to assess statistical significance in three non-overlapping time intervals (4–10, 10–16, and 16–22 h). To reduce noise, we applied a Gaussian filter to smooth the differentiation score trajectories. Next, we performed a two-sample $t$ test between the trajectories under different experimental conditions. Larger t-statistic indicates a lower likelihood of the observed differences arising by random chance. A parametric inference using RFT was conducted to determine the critical threshold with $\alpha = 0.05$, under the null hypothesis that there is no significant pattern in the observed data, namely, any difference between the cells subjected to distinct treatments is due to random chance. The resulting $P$ value represents the probability that the computed Gaussian fields would surpass the critical threshold and reject the null hypothesis (the null hypothesis is rejected if the $t$ value traverses the critical threshold). The RFT analysis was performed using the Python package rft1d in

the set level (the whole time interval), or in the cluster level in cases where the t test statistic field crossed the critical threshold.

## Correlation of differentiation score with time

The correlation between the single-cell differentiation scores and time was computed through the critical time interval where differentiation occurred (7.5–14.5 h). We used the Spearman correlation coefficient as a measurement for the monotonic increase in differentiation along a trajectory.

## Prediction of the onset of the differentiation process

The onset of the single-cell differentiation process was determined as the last stable time point below a threshold in the differentiation scores. This time point was defined as the last time point of the longest sequence with differentiation scores that ranged between values 0.2–0.3.

## Quantification of single cell predicted duration of the differentiation process

The *differentiation process duration* is a proxy for the time a single cell undergoes differentiation. The duration of the single-cell differentiation process was determined as the time passed from the predicted onset of differentiation to reaching a high stable threshold in the differentiation scores. The high stable threshold was defined as the first time point of the longest sequence with differentiation scores that ranged between 0.7–0.8. The differentiation process duration was calculated as the time passed between the low and high stable thresholds.

## Simple single-cell measurements and corresponding classifiers

We calculated single-cell time series of the following measurements:

- Local density: the number of nuclei within a radius of 50 μm around the cell.
- Speed: $speed(t) = \sqrt{(x_t - x_{t-1})^2 + (y_t - y_{t-1})^2}$, where $x_t$, $y_t$ are the nuclei $(x, y)$ position at time $t$.
- Mean actin intensity: mean actin intensity in a quantification window of $32 \times 32$ μm around the nuclei.
- Persistence: The ratio between a single cell's displacement and its full path length. Persistence of 1 implies that the cell migrated in a straight line.

For each measurement, and for all four together, we trained random forest classifiers with the mean value in each temporal segment to discriminate between undifferentiated and differentiated cells. We evaluated the discrimination performance of each of the five classifiers as described above.

## Quantification of single-cell terminal differentiation time

The *terminal differentiation time* of a single cell is an estimation based on the first time point of the longest sequence of differentiation scores that are higher than a threshold value of 0.78 (to avoid local peaks).

## Manual annotation of fusion events timing

In total, 68 nuclei from 6 fibers were backtracked to the frame when they fused into the fiber syncytium (Appendix Fig. S16).

## Statistical analysis

Pearson correlation (scipy.stats.pearsonr function) was used to assess the correlation between the terminal differentiation time and fusion since we assumed a linear correlation between them (Fig. 4D). Spearman correlation (using scipy.stats.pearsonr function) was used for correlating the monotonic increase in the differentiation trajectories with time. D'Agostino's K-squared test (using scipy.stats.normaltest) was used to determine the normality of distributions: duration of the differentiation process, terminal differentiation time, fusion time, and duration between differentiation and fusion.

# Data availability

Source code and sample data are publicly available, https://github.com/zaritskylab/muscle-formation-prediction. The mass spectrometry proteomics data have been deposited to the ProteomeXchange Consortium via the PRIDE (Perez-Riverol et al, 2022) partner repository with the dataset identifier PXD047198.

# Peer review information

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

## Acknowledgements

This research was supported by the Israeli Council for Higher Education (CHE) via the Data Science Research Center, Ben-Gurion University of the Negev, Israel (to AZ), and by the Wellcome Leap Delta Tissue program (to AZ). This project also received funding from the European Research Council (ERC StG # 851080 and ERC-PoC-2022 SuperFusion to OA). OA also acknowledges funding from the David Barton Center for Research on the Chemistry of Life and the Ruth and Herman Albert Scholarship Program for New Scientists. OA is an incumbent of the Miriam Berman Presidential Development Chair. We thank Shahar Golan for critically reading the manuscript. We thank Yoseph Addadi and the MICC Cell Observatory at the Weizmann Institute of Science for training and assistance.

## Author contributions

**Amit Shakarchy**: Data curation; Software; Formal analysis; Investigation; Methodology; Writing—original draft; Writing—review and editing. **Giulia Zarfati**: Data curation; Investigation; Writing—review and editing. **Adi Hazak**: Data curation; Investigation; Writing—review and editing. **Reut Mealem**: Data curation; Formal analysis; Investigation; Writing—review and editing. **Karina Huk**: Data curation. **Tamar Ziv**: Resources; Formal analysis; Investigation; Methodology; Writing—review and editing. **Ori Avinoam**: Conceptualization; Supervision; Funding acquisition; Investigation; Writing—original draft; Writing—review and editing. **Assaf Zaritsky**: Conceptualization; Supervision; Funding acquisition; Investigation; Writing—original draft; Writing—review and editing.

## Disclosure and competing interests statement

OA is a co-founder of ProFuse Technologies LTD. The remaining authors declare no competing interests.

# Expanded View Figures

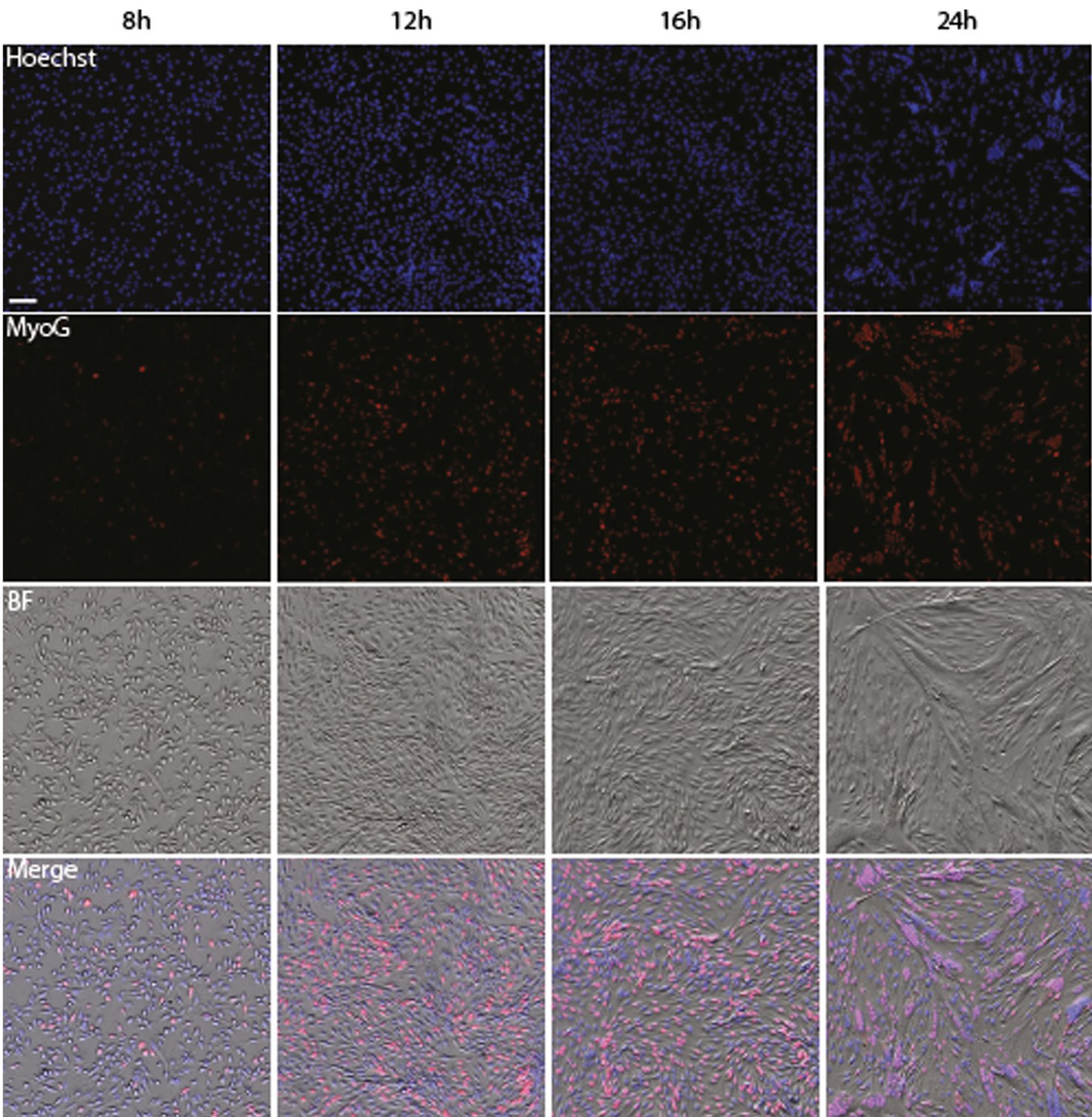

**Figure EV1.  The number of differentiating myoblasts MyoG-expressing increase over time.**

Images of primary myoblasts fixed at different time points after ERK inhibition and stained for nuclei (Hoechst, blue) and MyoG (red), along with Brightfield (gray) and Merge for reference. Magnification ×5. Scale bar 100 μm.

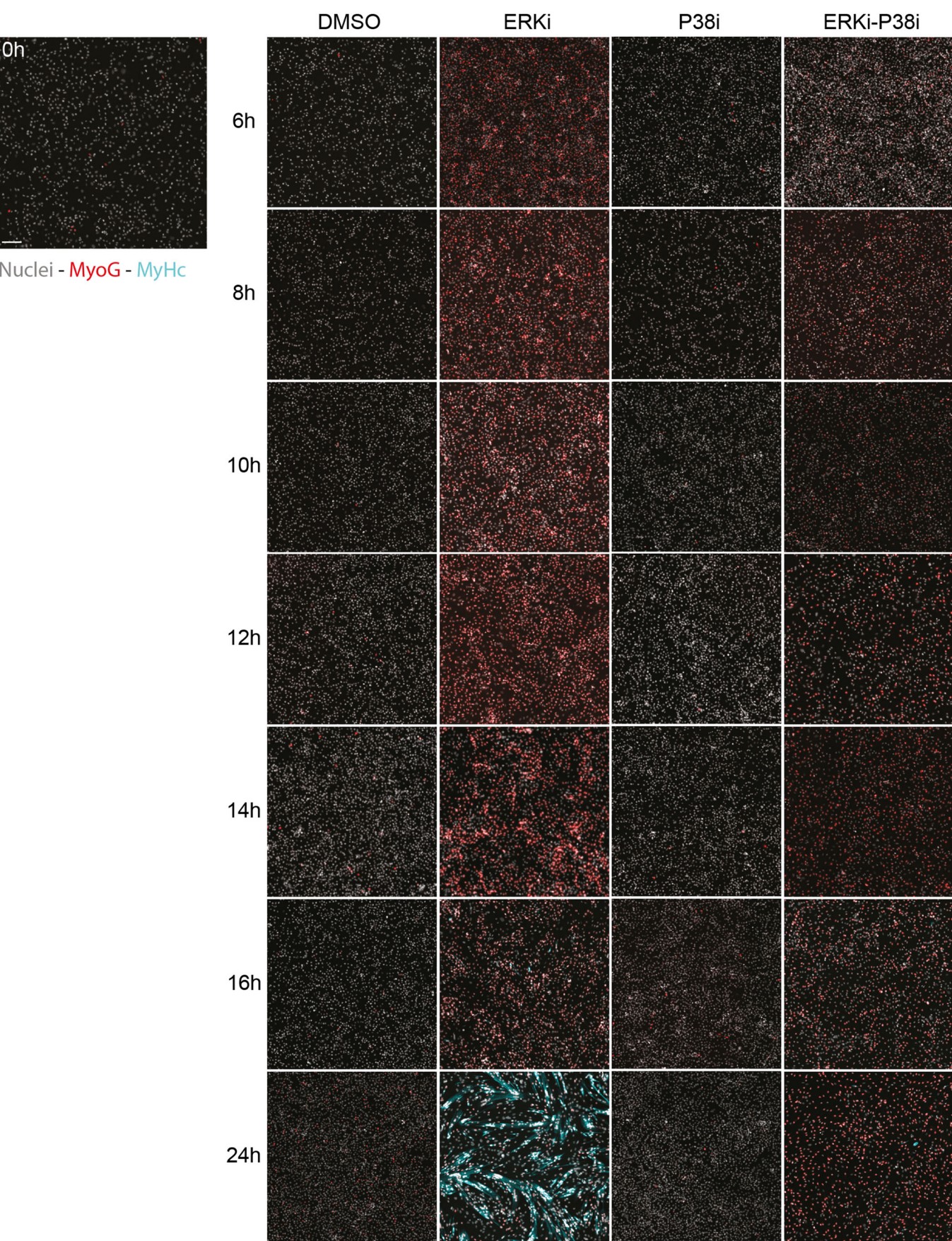

**Figure EV2.   Immunofluorescence staining of MyoG and MyHC.**

Representative immunofluorescence (IF) images of myoblasts at 0, 6, 8, 10, 12, 14, 16, 24 h after treatment with DMSO, p38i 5 µM or ERKi 1 µM or the combination of ERKi-p38i. Cells were stained using anti-MyoG (red), anti-MyHC (cyan), and the nuclear dye Hoechst 33342 (gray). Magnification ×5. Scale bar: 100 µm.

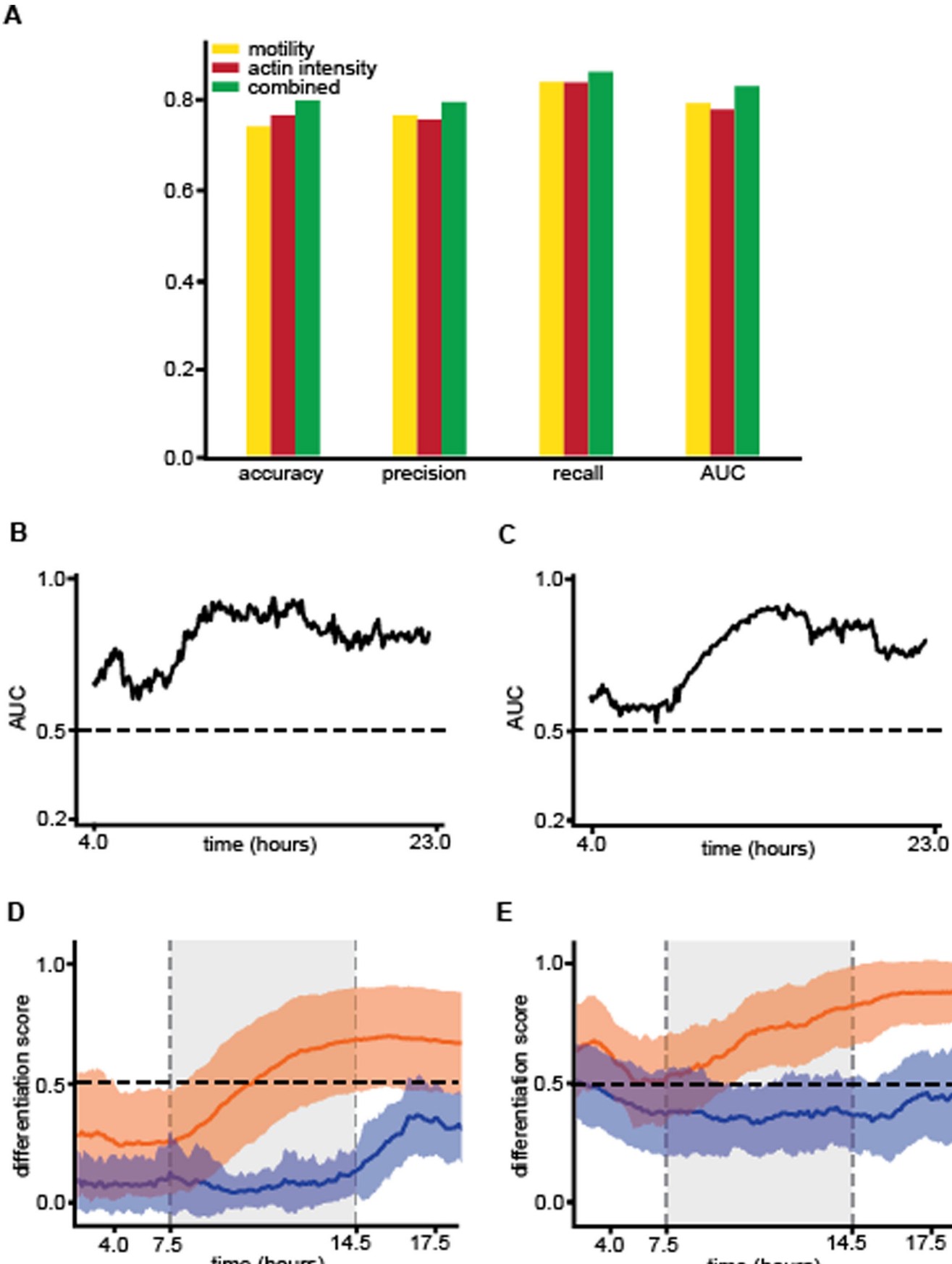

**Figure EV3.  Performance analysis of a classifier trained on both motility and actin dynamics.**

(A) Accuracy, precision, recall and area under the receiver operating characteristic (ROC) curve (AUC) for classifiers trained with motility (yellow), actin intensity (red) and a combination of motility and actin intensity (green) time series. Average accuracy rates were 0.74, 0.77 and 0.8; average precision rates were 0.77, 0.76, and 0.8; Average recall rates were 0.84, 0.84 and 0.87; average AUC rates were 0.8, 0.78 and 0.84 correspondingly. All metrics were calculated for 678 cells from an independent experiment. Overall, the combined classifier exhibits better classification performance. (B) Area under the receiver operating characteristic (ROC) curve (AUC) over time for a combined model (N=678 cells). (C) AUC over time for a combined model- flipped experiments for train/test (848 cells). (D) Mean (solid line) and standard deviation (shade) of the differentiation score over time of ERKi- (orange) and DMSO- (blue) treated cells using the combined classifier. Dashed vertical gray rectangle highlights the time interval of 7.5–14.5 h, where the model predicted the differentiation occurs (ERK: 575 cells; DMSO: 103 cells). (E) Mean (solid line) and standard deviation (shade) of the differentiation score over time of ERKi- (orange) and DMSO- (blue) treated cells using the combined classifier-flipped experiments for train/test (ERK: 538 cells; DMSO: 310 cells).

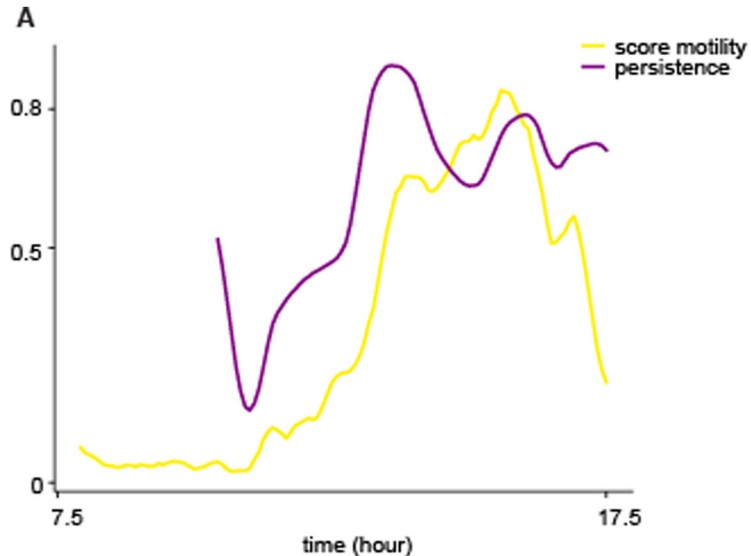

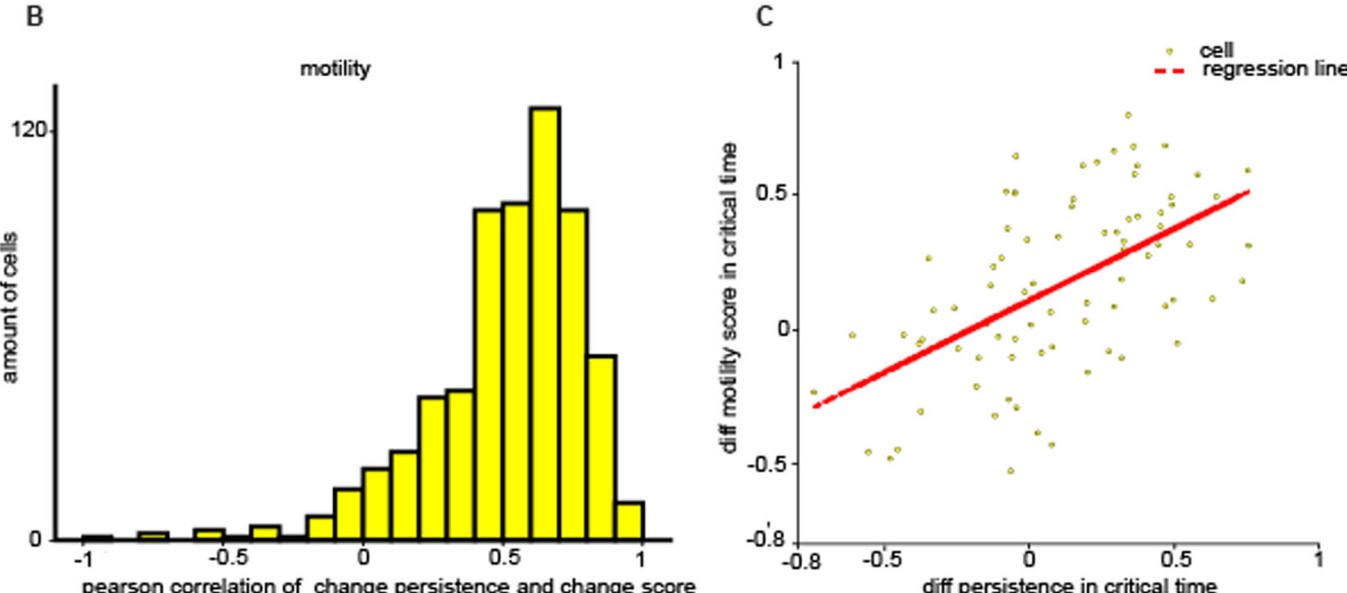

**Figure EV4. Persistence in migration is associated with differentiation.**

(A) Representative single cell's differentiation score (yellow) as predicted by the motility-based model, and persistence in migration rate (purple) through time. (B) Distribution of single cells Pearson correlation between the difference in differentiation scores and the difference in persistence rates. Values were calculated within time intervals of 50 min. Mean Pearson correlation coefficient was 0.51. 88.07% of cells showed significant Pearson correlation (Pearson correlation $P$ value < 0.05; $N = 575$ cells). (C) Single cells difference in differentiation scores ($y$ axis) over difference in the persistence rate ($x$ axis) between the beginning and end of the critical time window where we identified differentiation occurs (7.5–14.5 h). Red diagonal line indicates the regression line. Pearson correlation coefficient was 0.55 (Pearson correlation $P$ value < 0.0001).

Before Washing        After Washing

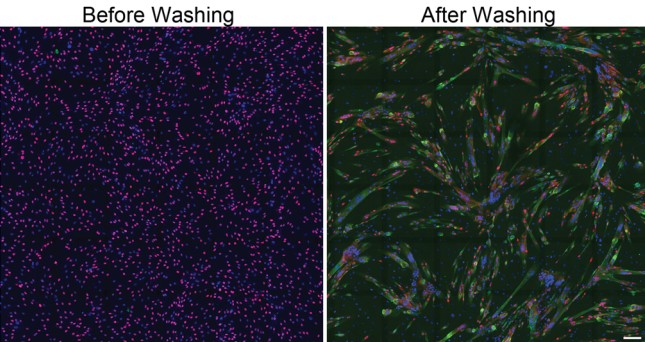

**Figure EV5.   Co-inhibition of p38 and ERK1/2 leads to properly differentiated ready to fuse cells.**

Immunofluorescence images of primary myoblasts treated with p38i+ERKi for 24 h, before and 24 h after the inhibitors were washed. The cells were fixed and stained for MyoG (red), MyHC (green), and the nuclei (Hoechst, blue), scale bar 100 µm. 24 h after treatment with p38i+ERKi, most of the cells expressed MyoG, were MyHC negative, and did not undergo fusion. Thus, the cells were differentiated but unfused. 24 h after the inhibitors were washed, there was a decrease in MyoG-positive cells, and most cells express MyHC, indicating that cells completed the differentiation process and fused.

