## [Peer Review File · Molecular Systems Biology]

Machine learning inference of continuous cell state transitions during myoblast differentiation

AmitShakarchy,Giuliazarfati,AdiHazak,ReutMealem,KarinaHuk,TamarZiv,OriAvinoam,andAssafZaritsky

DOI: 10.15252/msb.202311631

Corresponding author(s): Ori Avinoam (ori.avinoam@weizmann.ac.il), Assaf Zaritsky (assafzar@gmail.com)

Review Timeline:

Submission Date:	2nd Mar 23
Editorial Decision:	12th Apr 23
Revision Received:	19th Oct 23
Editorial Decision:	21st Nov 23
Revision Received:	19th Dec 23
Accepted:	4th Jan 24

Editor: Maria Polychronidou

Transaction Report:

12th Apr 2023

RE: Manuscript MSB-2023-11631, Machine learning inference of continuous single-cell state transitions during myoblast differentiation and fusion

Dear Assaf,

Thank you again for submitting your work to Molecular Systems Biology. We have now heard back from the three referees who agreed to evaluate your study. As you will see below, the reviewers raise substantial concerns on your work, which unfortunately preclude its publication in Molecular Systems Biology.

The reviewers appreciate that the study seems potentially interesting. However, they are not convinced that the main conclusions are well supported and mention substantial limitations both from a methodological and a biological point of view. As such, the reviewers indicated that they do not support publication in Molecular Systems Biology.

Taken together and given the substantial concerns raised by the reviewers, I am afraid I see no other choice than to return the manuscript with the message that we cannot offer to publish it. I am sorry that the review of your work did not result in a more favorable outcome on this occasion, but I hope that you will not be discouraged from submitting future work to Molecular Systems Biology. In any case, thank you for the opportunity to examine this work.

Kind regards,

Maria

Maria Polychronidou, PhD
Senior Editor
Molecular Systems Biology

Reviewer #1:

Shakarchy et al. demonstrate a machine learning-based approach to quantify single-cell myoblast differentiation. Using an ML model trained on single-cell time series acquired with live cell imaging, they can calculate a differentiation score that can predict the differentiation state of individual myoblasts. The paper is clearly written and contains a well-documented methodology, although notebooks with the analysis should be provided with the publication (as promised by the authors). The experimental validation with p38 inhibition is a valuable addition. This work can serve as a resource for quantitative biologists who would like to perform similar analyses from imaging data.

However, my main objection to recommending this work for publication in MSB is the lack of methodological or biological novelty that would be required for this journal. In my opinion, this work would be more suitable for a more technical, methods-type journal. Data analysis is pretty standard and relies on extracting features from time series and using them to train random forest classifiers. The observation that the model trained on combined features of actin intensity can predict the differentiation state from myoblasts to myocytes is hardly surprising, since organised actin filaments are the hallmark of muscle cells.

Minor points include:

1. What was the rationale for partitioning the time series into 2.5-hour-long windows? Why this length?
2. Have the authors considered using information about the neighbours to improve the classification performance?

Reviewer #2:

The authors of the study proposed a novel approach to track the intermediates states of cells during differentiation, which can be challenging due to changes in their internal states. They introduced a quantitative metric, based on the confidence score from the Random Forest classifier, to monitor the continuous differentiation state of single myoblasts over time. This approach revealed that inhibiting ERK1/2 leads to a gradual transition from an undifferentiated to a terminally differentiated state 7.5-14.5 hours post-inhibition, and myoblast fusion occurred approximately three hours after predicted terminal differentiation.

While the proposed differentiation score is an important concept, there are several points that need further clarification. The authors trained the classifier using motility or actin intensity, and found that the differentiation score can be useful for this purpose. However, when they used different features to train the classifiers, the scores were not sufficiently useful, and they did not provide enough information about the characteristics of their supervised learning that made their differentiation score have desirable features. Thus, to make their approach generally applicable, these characteristics need to be addressed. Furthermore, the authors showed that their differentiation score correlated with differentiation markers on the population level. However, it is not clear whether this correlation will hold true at the single cell level. While there exist some cell populations with high Spearman correlation coefficients, there are significant portions of cells with low Spearman correlation, and thus it is not clear how reliable the differentiation score will be as a readout at the single cell level.

The following specific points need to be addressed in more detail:

1. The information about the classifier training is insufficient, as the authors only presented AUC and did not provide other measures such as accuracy, sensitivity, specificity, etc.
2. There are two differentiation scores from motility and actin intensity, but the authors did not provide sufficient explanation and discussion about the differences between them. Also, it is also not clear if combining motility and actin intensity to train the classifiers could produce better differentiation score.
3. The large standard deviation in Fig. 1 C,D makes it unclear whether the changes are statistically significant, and the authors should perform statistical testing. Plotting how much change happened in comparison to the initial time point for individual cells during differentiation could also be helpful.
4. There may be labeling errors in Fig. 2I that need to be corrected.
5. The interpretability of the study needs to be better integrated throughout.
6. While it is good to show the variability of the data using standard deviations in the time course plots, the authors could also include 95% CI or SEM to better illustrate statistical significance.
7. The authors claim that there is a coupling between differentiation and fusion because cells undergo fusion within a typical time interval from their terminal differentiation. However, it is unclear why this suggests coupling, as the same thing can happen without it. The same applies to the correlation plots.
8. In Fig. 5E-F, the differentiation scores from actin intensity are significantly lower than those from motility, suggesting that the differentiation score from actin intensity may have some limitations. The authors should discuss this issue in more detail.

Reviewer #3:

In this paper, Shakarchy et al. develop a machine learning model to measure myoblast differentiation using measurements of motility and actin intensity from time-lapse imaging experiments. The authors use an innovative combination of high-dimensional feature extraction from time series data and interpretable machine learning to provide a quantitative description of continuous cell state changes at the single-cell level - deriving a differentiation "score" based on the certainty measurements of the model itself. Using this method, the authors claim that ERK1/2 inhibition leads to a gradual transition from undifferentiated to differentiated myoblasts 7.5-14.5h post treatment and that their model predicts terminal differentiation even in conditions where subsequent fusion is inhibited.

While I believe that this is an exciting method that could be very useful in addressing similar questions in other biological contexts, I am not convinced that the claims made by the authors about

the biology in question are adequately supported by the data/analyses that are presented. I am not an expert in myoblast differentiation, so I cannot assess the impact of this work in that field, nor how well the data presented aligns with what is already known. However, I would like to see some additional experiments/analyses to support some specific claims made by the authors about the single-cell behavior they observed.

Major Points

A main concern in this manuscript is the description of heterogeneity at the single-cell level. The authors claim in the introduction that myoblast differentiation is difficult to study because of the complex heterogeneity that exists. While their model is trained on single-cell data and is used to predict the differentiation progress of individual cells, there is very little interrogation of the extent or role of heterogeneity in this process. Most figures present time traces as means \pm -SD; however, these can obscure some important single-cell behaviors. The authors address this question: saying that the gradual increase that they detect at the population level (Fig 2D-E) could be explained by either (1) synchronized gradual transitions of single-cells, or (2) unsynchronized abrupt transitions. They say that they mostly see gradual increases in differentiation scores but abrupt transitions "were not observed". Perhaps this is a typo because, in the single cell traces shown in Fig 2G, there are clear differences in the slopes of the scores - with the blue curve in the upper plot certainly looking quite abrupt compared to the other traces. Fig 2H also shows that the duration of this increase in score varies substantially among cells, as noted by the authors. Therefore, I am not sure how the conclusion is reached that the "results supported the former mechanisms of synchronized and gradual-continuous transition<s>". Fig 2G clearly shows that the transitions are not synchronous, and cells certainly vary in the speed of their transitions. The authors should compare the starting times of these transitions across all cells to test their claim of synchronicity. It would also be fine to report if cells exhibited heterogeneity in the dynamics of this transition (in terms of timing and speed), in which case the authors should amend their conclusions here.

I also do not think that the validation experiment in Fig 2I is appropriate to test if the transition is gradual. These results are consistent with the time frame over which the transitions are predicted by the model to occur (8.5-14.5h), but these results could also be obtained if cells abruptly transitioned at different time points.

The authors also claim that simple single-cell measurements are insufficient to quantify cell state transitions (Fig 3). Why not fold these measurements into the model? Couldn't the feature extraction of motility/actin dynamics over specific time windows be combined with these simple measures at timepoints in the middle of these windows? In fact, why present the actin and motility models separately? Wouldn't combining all of these measurements produce an even more accurate model?

In the final experiment to uncouple differentiation from fusion, the authors claim that their model can predict differentiation even when fusion is inhibited using a p38 inhibitor in combination with their ERK inhibitor. There are clearly differences in the dynamics of MyoG gain (Fig 5C) and MyoD loss (Fig 5D). In addition, the actin model performs worse in the p38i/ERKi condition; however, it is difficult to assess if the differences in these curves is significant since the authors do not show variance. Even so, the actin model maxes out at a model certainty value of 0.5, which by definition cannot distinguish undifferentiated from differentiated cells.

** As a service to authors, EMBO Press offers the possibility to directly transfer declined manuscripts to another EMBO Press title or to the open access journal Life Science Alliance launched in partnership between EMBO Press, Rockefeller University Press and Cold Spring Harbor Laboratory Press. The full manuscript and if applicable, reviewers' reports, are automatically sent to the receiving journal to allow for fast handling and a prompt decision on your manuscript. For more details of this service, and to transfer your manuscript please click on <https://msb.msubmit.net/cgi-bin/main.plex?el=A1k6xow6A2CmwY2X7A9ftd4wFrv0wBfeBVF9t4Qc847gY>. **

Assaf Zaritsky, Ph.D.
Assistant Professor

Faculty of Engineering Sciences
Software and Information Systems Engineering

October 19th, 2023

To: Dr. Maria Polychronidou, Senior Editor, *Molecular Systems Biology*

Point-by-point response

We appreciate the feedback provided by all the reviewers, acknowledging the potential interest in our study but expressed concerns about the methodological and biological novelty. We have followed the revision plan that was agreed-upon with the editor. Specifically, we incorporated additional proteomics data and analyses that we believe addresses the concerns raised by the reviewers and that significantly enhance the novelty, innovation, and impact of our work. These are detailed in the following point-by-point response.

Reviewer #1:

Shakarchy et al. demonstrate a machine learning-based approach to quantify single-cell myoblast differentiation. Using an ML model trained on single-cell time series acquired with live cell imaging, they can calculate a differentiation score that can predict the differentiation state of individual myoblasts. The paper is clearly written and contains a well-documented methodology, although notebooks with the analysis should be provided with the publication (as promised by the authors). The experimental validation with p38 inhibition is a valuable addition. This work can serve as a resource for quantitative biologists who would like to perform similar analyses from imaging data.

The code and example data are available via a notebook,
<https://github.com/amitshakarchy/muscle-formation-regeneration>.

However, my main objection to recommending this work for publication in MSB is the lack of methodological or biological novelty that would be required for this journal. In my opinion, this work would be more suitable for a more technical, methods-type

journal. Data analysis is pretty standard and relies on extracting features from time series and using them to train random forest classifiers.

We respectfully disagree with the notion of a lack of methodological and biological novelty, and would like to highlight the unique aspects of our approach, as well as the distinct biological insights we have gained through our study:

Methodological and conceptual novelty:

The novelty of our work does not primarily lie in the extraction of temporal features or the use of random forest classifiers. Instead, our study presents a conceptual innovation in utilizing the **single cell classifier score as a continuous readout to measure a biological process that evolves over time**. Our study is the first to demonstrate measuring single-cell continuous state transitions using supervised machine learning applied to time series data extracted from live cell imaging. This approach overcomes the limitations of previous methods that relied on computational construction of "pseudo-time" trajectories from fixed images, manual annotation of intermediate cell states, or unsupervised representations. In the manuscript, we propose the hypothesis that the classifier score can serve as a continuous readout and proceed to validate this hypothesis and explore its potential for uncovering novel biological insights. This is particularly challenging in a process that lacks real-time markers for validation, but addressing this challenge is crucial, especially as similar approaches can be applied to other cellular processes without live "state" markers.

We also present the first evidence of correlating a computationally derived single-cell state readout with independently measured functional readouts, namely, differentiation to fusion, and in the revision, cell migration persistence to differentiation state. This suggests that our approach could be a powerful tool for revealing connections between dynamic single-cell states and practically any other single cell functional readout, enabling systematic characterization of the full spectrum of heterogeneities in complex biological processes. We envision broad applications of similar inference of continuous cell state in other biological

processes, such as cell cycle, epithelial-to-mesenchymal transition, immotile-to-motile, disease progression, immune cell activation and cell death.

To our knowledge, this work is the first to explicitly and systematically highlight the importance of using temporal information to quantitatively characterize continuous cell states in dynamic processes. Our manuscript, therefore goes beyond most method papers by deriving and validating new biological insights through computational-experimental iterations. We further strengthened these insights in the revised manuscript with additional data and analyses (as detailed below).

Biological novelty:

Compared to other studies that develop methods to measure cell states, mostly focusing on the well characterized process of cell cycle, we focus on the much less-studied and much more challenging and physiologically relevant process of cellular differentiation in the unique setting of skeletal muscle cells that become multinucleated fibers, which is characterized by two known discrete cell states with unknown intermediate states and no readily available live cell markers to monitor the process.

We were able to quantitatively follow this process via live imaging thanks to a unique experimental setting that allows us to induce robust and synchronous initiation of differentiation by inhibiting the ERK1/2 signaling pathway (Eigler et al. 2021., visualization in a new Fig EV1). We concluded that differentiation occurs gradually 7.5-14.5 hours after ERK1/2 inhibition and that fusion initiates approximately 3 hours after differentiation. Moreover, we identified experimental conditions to uncouple the differentiation and fusion processes, allowing us to study these two events independently. Importantly, these results were consistent for two independent models trained with different temporal readouts (motility, actin). Taken together these results address a long standing question in the field about the temporal relationship between differentiation and fusion, suggesting that the two events are sequentially coordinated but independently regulated. This has the potential to

deepen our understanding of the underlying mechanisms and regulatory factors in myoblast differentiation and fusion, representing a significant advancement in the field of muscle biology. In addition, for the revision, we acquired mass spectrometry data under these distinct conditions, to further validate that differentiation indeed occurs while the fusion machinery is specifically not expressed. This analysis demonstrated that fusion is directly regulated by p38 but also points towards novel putative fusion-related proteins, following a specific prediction of our computational model, thereby providing new insights into the molecular players involved in this process that are included in the revised manuscript. In summary, our approach not only offers a more comprehensive and accurate understanding of the cellular differentiation and fusion processes, but also paves the way for new insights in the field.

While our writing primarily emphasized the robustness of our method, we acknowledge that we did not provide a comprehensive review of the literature pertaining to myoblast differentiation and its associated limitations. In the revision, we delved deeper into the current state of the field, including a more thorough examination of the advantages of using ERK inhibition as a means of inducing differentiation. In light of these considerations, we believe that our study offers both methodological and biological novelty that would be of interest to the broad readership of MSB.

New Figure EV1:

The observation that the model trained on combined features of actin intensity can predict the differentiation state from myoblasts to myocytes is hardly surprising, since organised actin filaments are the hallmark of muscle cells.

We agree with the reviewer that selecting actin as a feature was a rational choice; however, our model's ability to predict the differentiation state from myoblasts to myocytes using actin intensity features showcases the novelty and effectiveness of our approach, rather than being an expected outcome based solely on the known role of actin in muscle cells.

First, it is important to emphasize that while myoblasts begin expressing muscle-specific actin isoforms and forming nascent myofibrils as they exit the cell cycle, these early structures do not mature into sarcomeres until after cells fuse and myotube maturation begins. To our knowledge, there is no evidence in the literature suggesting that the process of nascent myofibril formation can be a surrogate for the differentiation state. Moreover, as myoblasts differentiate, they elongate, change their motility and align with one another, processes also driven by actin dynamics. Considering these factors, it becomes evident that the relationship between actin intensity features and the differentiation state is complex and not trivial to correlate. Moreover, we specifically demonstrate that the mean actin intensity over time can partially distinguish between differentiated and undifferentiated cells (Fig 3F) but cannot provide a continuous readout of the differentiation state (Fig 3G). In the revised manuscript, we further substantiated this claim by assessing the temporal derivative in single cell actin intensity as a sole readout (new Fig 3D, and adjustments of Fig 3F-G), demonstrating that the relationship between features of actin intensity and the differentiation state is not trivial.

Revised Figure 3:

Minor points include:

1. What was the rationale for partitioning the time series into 2.5-hour-long windows?

Why this length?

We selected a temporal segment size of 2.5 hours because of high performance with respect to shorter time-segments as shown in Appendix Fig S5 (“Classification sensitivity analysis: temporal segment size”) on the effect of the time window on the model’s performance. The shorter time frame was preferable because the exact time of differentiation was not known a priori and was only roughly approximated according to the first fusion event, and thus longer times could label cells when they are not yet terminally differentiated.

2. Have the authors considered using information about the neighbours to improve the classification performance?

We calculated the correlation between differentiation trajectories of pairs of close- versus far-cell pairs, but could not find any evidence for spatial patterns of similarity in differentiation progression. Hence, we did not report these data in the manuscript. For the revision we tested whether myoblasts that fused to the same myofiber had more similar differentiation trajectories than myoblasts that fused to other myofibers, but could not find any supportive evidence for this hypothesis.

Reviewer #2:

The authors of the study proposed a novel approach to track the intermediates states of cells during differentiation, which can be challenging due to changes in their internal states. They introduced a quantitative metric, based on the confidence score from the Random Forest classifier, to monitor the continuous differentiation state of single myoblasts over time. This approach revealed that inhibiting ERK1/2 leads to a gradual transition from an undifferentiated to a terminally differentiated state 7.5-14.5 hours post-inhibition, and myoblast fusion occurred approximately three hours after predicted terminal differentiation.

While the proposed differentiation score is an important concept, there are several points that need further clarification. The authors trained the classifier using motility or actin intensity, and found that the differentiation score can be useful for this purpose. However, when they used different features to train the classifiers, the scores were not sufficiently useful, and they did not provide enough information about the characteristics of their supervised learning that made their differentiation score have desirable features. Thus, to make their approach generally applicable, these characteristics need to be addressed.

We thank the reviewer for appreciating the conceptual advance in our manuscript. In the revised manuscript we elaborated on the justifications for what features were important to enable quantitative description of the differentiation process. The key idea is selecting the properties that are expected to change along the progression of the continuous process and extracting rich quantitative descriptions of their dynamics.

We show in Fig 1 that changes in actin and motility correlate with differentiation at the population level, and this was our justification for selecting (single cell) actin/motility dynamics as the input readout for our model as stated in page 7 lines #138-146: *“Following the association between differentiation and the population scale changes in actin intensity and motility,*

we hypothesized that the information encoded in single-cell migration trajectories and actin dynamics might be sufficient to computationally estimate a continuous score reflecting a myoblast's gradual transition from an undifferentiated proliferative state to a terminally differentiated fusion competent state. To test this hypothesis, we took a machine learning approach: (1) extracting features from the motility/actin time-series, (2) training machine learning classification models (aka classifiers) to discriminate between the undifferentiated and differentiated states, and (3) using the confidence of these models as a quantitative measurement for cell state."

In the revision, in a new Appendix Fig S1 we explicitly present the actin/speed at the single cell level after tracking as a motivation for using these measures for our predictive model, page 7 lines #156158: *"Single cell analysis confirmed our population-based results that differentiation was accompanied by a decrease in cell motility and an increase in the F-actin marker's fluorescence intensity (Appendix Fig S1)."*

In contrast, local cell density, which also changes over time, but before the onset of differentiation, was able to discriminate between differentiated and undifferentiated cells (Fig 3F), but was not correlated with the differentiation process (Fig 3G).

We defined monotonicity as a desired property of the confidence score over time (Fig 2), which highlighted the importance of temporal information to measure a continuous process (Fig 3). We reflected on that on page 11, lines #280-283: *"In our case, and in agreement with other studies (Copperman et al., 2021; Wang et al., 2020; Wu et al., 2022), integration of multiple dynamic features encoding the temporal changes were necessary to continuously measure a biological process."*, and in the Discussion.

Appendix Figure S1:

Furthermore, the authors showed that their differentiation score correlated with differentiation markers on the population level. However, it is not clear whether this correlation will hold true at the single cell level. While there exist some cell populations with high Spearman correlation coefficients, there are significant portions of cells with low Spearman correlation, and thus it is not clear how reliable the differentiation score will be as a readout at the single cell level.

We agree with the reviewer that the main challenge of our manuscript is validating that the differentiation score is a reliable readout at the single cell level. This is hard because there are no “ground truth” differentiation markers that are suitable for live imaging. This is exactly the experimental limitation that our approach was designed to tackle.

In most cells the correlation between time and the differentiation score was high. In Fig. 2F and in Fig 3G we show that for the motility-based classifier the median correlation is 0.49, and for the actin-based classifier the median correlation is 0.69. While it is expected that for such a complex and heterogeneous process there will not be a perfect agreement, we agree that further support is needed to convince the readers that this is a truth worthy readout for the single cell differentiation state.

In the original version of the manuscript we tackled this challenge by:

1. Demonstrating agreement at the single cell level between the motility- and actin-based classifiers' predictions, two independent classifiers that were trained on distinct cell functional readouts showing high correlation beyond 0.5 for 62% of cells, and negative correlation for < 4% (Appendix Fig S10). This is now stated explicitly in page 9 lines #214-217: *“The motility- and actin-based classifiers' predictions were mostly consistent at their single cell predictions, showing high correlation beyond 0.5 for 62% of cells, and negative correlation for less than 4%, and providing further support that both models measure the continuous state transition (Appendix Fig S10A-B).”*
2. Demonstrating that static measurements can discriminate between differentiated and undifferentiated cells but without providing a monotonic prediction that is indicative of a continuous process (Fig 2F, Fig 3G - see previous point).
3. High correlation between terminal differentiation and fusion time (Fig 4).

In the revised version we demonstrated that the motility-based model was associated with persistent cell migration. This finding links our single cell differentiation score to a single cell functional readout, providing a concrete demonstration how the (not perfect) single cell differentiation score can be used as a readout at the single cell level to derive new biological insights. These results are now reported in a new Fig EV4 and in page 9, lines #222-230: *“High agreement between the two models was associated with the single cells’ motility persistence, the ratio between the direct translation (i.e., distance from start to end) and the overall distance traveled (Appendix Fig S10C). Qualitative and quantitative association between single cells’ motility persistence and motility-based (but not actin-based) differentiation score identified persistent motility as a functional marker for the intermediate states of myoblast differentiation (Fig EV4). The lack of association between the actin model and persistence suggests that the actin model encodes different dynamic properties that are linked to the differentiation. Moreover, this lack of association highlights the potential to use deviations between these models to discover mechanisms that uncouple the link between motility, actin dynamics and myoblast differentiation.”*. This demonstration strengthens the feasibility and further highlights the potential of using a continuous cell state readout.

Moreover, this latter point regarding scenarios where the models disagree, is capitalized in the revision using the single cell differentiation score to make a specific prediction regarding alteration of actin regulators (Fig 5). This prediction is confirmed via mass spectrometry analysis in a new Fig 6 (see later).

New Figure EV4:

The following specific points need to be addressed in more detail:

1. The information about the classifier training is insufficient, as the authors only presented AUC and did not provide other measures such as accuracy, sensitivity, specificity, etc.

In the revised manuscript we provided accuracy, sensitivity and specificity, in addition to the AUC. These results are now reported in a new Fig EV3 in the context of evaluating a joint actin-motility classifier (see below).

2. There are two differentiation scores from motility and actin intensity, but the authors did not provide sufficient explanation and discussion about the differences between them.

We thank the reviewer for pointing us to further explore the disagreement between the models. This made a pivotal impact on the revision. Specifically, we took a closer look at the disagreement between the models beyond what is shown in (the old) Fig S13, qualitatively and quantitatively examined the properties of cells where the models agreed or disagreed on in a new Appendix Fig S11 (qualitative) and a new Appendix Fig S10 (quantitative). Page 9 lines #217-222: *“Analysis of cells sub-groups partitioned according to the agreement between the motility and actin models, showed that lower agreement between the classifiers was associated with lower monotonicity of the differentiation scores of both models (Appendix Fig S10C). In most cases of disagreement between the motility and actin models, we noticed a deviation in the actin-based model when the cell entered a crowded region and/or crawled below other cells (Appendix Fig S11).”*.

This analysis revealed that motility persistence was associated with an agreement between the actin-based and motility-based models, and later used to link persistence to differentiation (new Fig EV4, see previous point). Lastly, disagreement between the motility-based and actin-based models raised the hypothesis regarding alteration in the actin machinery that was validated in a new Fig 6 (details below).

New Appendix Figure S11:

New Appendix Figure S10:

Also, it is also not clear if combining motility and actin intensity to train the classifiers could produce better differentiation score.

In the revised manuscript we reported the performance of a model trained on features extracted from both motility and actin intensity as the reviewer (and reviewer #3) suggested. Indeed, the combined classifier surpassed each of the motility/actin classifiers, and these results are now reported in a new Fig EV3, on page 8, lines #175-177: *“A classifier trained on features derived from both motility and actin time-series surpassed each of the motility/actin classifiers, suggesting that motility and actin dynamics contain complementary information regarding the cells’ state (Fig EV3A-C).”*

We would like to emphasize that the manuscript is not focused on what is the best set of features to measure a score following a continuous process, rather, the main point is demonstrating that our approach of using the confidence score of a classifier trained for a binary classification task can be used to continuously measure a biological process that evolves over time from one cell state to another. This is now described in page 9, lines #194-199: *“For the rest of the manuscript we focused on analyzing the motility- and actin-based models, because showing that each of two independent models trained with different temporal readouts (motility, actin) can quantitatively monitor the cell differentiation process, and thus strengthening our goal of evaluating whether the confidence score of a classifier trained for a binary classification task can be used to continuously measure a biological process that evolves over time.”*

New Figure EV3:

3. The large standard deviation in Fig. 1 C,D makes it unclear whether the changes are statistically significant, and the authors should perform statistical testing. Plotting how much change happened in comparison to the initial time point for individual cells during differentiation could also be helpful.

In a new Appendix Fig S8 we analyzed the temporal change in the single cells' differentiation score, its integration over time (i.e., comparison to the initial time point (instead of the raw actin/motility), and its statistical significance. Page 9 lines #208-213: *“Measuring the distribution of the per-cell differentiation scores' temporal derivative (Appendix Fig S8A-B), their integration over time (Appendix Fig S8C-F), the predicted onset (Appendix Fig S9), and the predicted duration (Fig 2H) of the differentiation process, suggested that the progression in single-cell differentiation is highly heterogeneous (Fig 2H). These results suggest a heterogeneous gradual transition from an undifferentiated to a differentiated state within a typical timeframe.”*

New Appendix Figure S8:

4. There may be labeling errors in Fig. 2I that need to be corrected.

We apologize. This figure panel, in its revised form, appears now in Fig 1B.

5. The interpretability of the study needs to be better integrated throughout.

We took better care to provide clear interpretation of the results throughout. We hope that the reviewer will agree.

6. While it is good to show the variability of the data using standard deviations in the time course plots, the authors could also include 95% CI or SEM to better illustrate statistical significance.

We performed multiple independent rounds of training and reported their consistency on page 9, lines #203-206: *“A similar gradual increase in differentiation score at 7.5-14.5 was observed*

when flipping the experiments used for training and testing (Appendix Fig S4), the differentiation score was not sensitive to the size of the temporal segment (Appendix Fig S5), nor to the window size used to measure actin (Appendix Fig S6), and was consistent across multiple independent trainings (Appendix Fig S7).“.

New Appendix Figure S7:

7. The authors claim that there is a coupling between differentiation and fusion because cells undergo fusion within a typical time interval from their terminal differentiation. However, it is unclear why this suggests coupling, as the same thing can happen without it. The same applies to the correlation plots.

We would like to clarify that we used the term "temporal coupling" early (Figure 4) in our manuscript because later, in Figure 5, we demonstrated the ability to decouple these processes through perturbation. However, to avoid confusion, we modified our terminology and instead suggested that differentiation and fusion were interdependent. We hypothesize that there is a differentiation checkpoint that must be reached before fusion can proceed, ensuring a clearer

and more accurate representation of our findings, page 12, lines #296-297: “These results suggest that myoblasts must reach a differentiation checkpoint before fusion can proceed.”.

8. In Fig. 5E-F, the differentiation scores from actin intensity are significantly lower than those from motility, suggesting that the differentiation score from actin intensity may have some limitations. The authors should discuss this issue in more detail.

We thank the reviewer for focusing our attention on this result, because this P38+ERKi induced alteration in the actin model turned out to be a discovery! Following this observation, we collected mass spectrometry data at these conditions. This analysis revealed that (1) cells were differentiated upon these conditions, (2) regulators of the actin cytoskeleton were differentially expressed, suggesting that our model was able to predict these alterations. These results are now described in a new Figure 6 and are extensively discussed in the text.

New Figure 6:

Reviewer #3:

In this paper, Shakarchy et al. develop a machine learning model to measure myoblast differentiation using measurements of motility and actin intensity from time-lapse imaging experiments. The authors use an innovative combination of high-dimensional feature extraction from time series data and interpretable machine learning to provide a quantitative description of continuous cell state changes at the single-cell level - deriving a differentiation "score" based on the certainty measurements of the model itself. Using this method, the authors claim that ERK1/2 inhibition leads to a gradual transition from undifferentiated to differentiated myoblasts 7.5-14.5h post treatment and that their model predicts terminal differentiation even in conditions where subsequent fusion is inhibited.

While I believe that this is an exciting method that could be very useful in addressing similar questions in other biological contexts, I am not convinced that the claims made by the authors about the biology in question are adequately supported by the data/analyses that are presented. I am not an expert in myoblast differentiation, so I cannot assess the impact of this work in that field, nor how well the data presented aligns with what is already known. However, I would like to see some additional experiments/analyses to support some specific claims made by the authors about the single-cell behavior they observed.

We appreciate the reviewer's recognition of the potential utility of our method in various biological contexts. We performed additional experiments and analyses to substantiate the claims made about the single-cell behavior observed during myoblast differentiation as detailed below. We also conducted a more thorough review of the relevant literature regarding actin and motility during differentiation to better contextualize our findings within the field.

Major Points

A main concern in this manuscript is the description of heterogeneity at the single-cell level. The authors claim in the introduction that myoblast differentiation is difficult to study because of the complex heterogeneity that exists.

The argument regarding heterogeneity, in this context, was supposed to imply that proliferating myoblasts differentiate on a much longer and variable time scale in respect to ERK inhibition. Thus, ERK inhibition enables us to study this process because differentiation initiates earlier and is less variable (but still heterogeneous). We agreed with the reviewer that this may be confusing and thus avoided using the term “synchronous”, and instead used “faster” and “less temporally variable”.

While their model is trained on single-cell data and is used to predict the differentiation progress of individual cells, there is very little interrogation of the extent or role of heterogeneity in this process. Most figures present time traces as means \pm SD; however, these can obscure some important single-cell behaviors. The authors address this question: saying that the gradual increase that they detect at the population level (Fig 2D-E) could be explained by either (1) synchronized gradual transitions of single-cells, or (2) unsynchronized abrupt transitions. They say that they mostly see gradual increases in differentiation scores but abrupt transitions "were not observed". Perhaps this is a typo because, in the single cell traces shown in Fig 2G, there are clear differences in the slopes of the scores - with the blue curve in the upper plot certainly looking quite abrupt compared to the other traces. Fig 2H also shows that the duration of this increase in score varies substantially among cells, as noted by the authors. Therefore, I am not sure how the conclusion is reached that the "results supported the former mechanisms of synchronized and gradual-continuous transition<s>". Fig 2G clearly shows that the transitions are not synchronous, and cells certainly vary in the speed of their transitions. The authors should compare the starting times of these

transitions across all cells to test their claim of synchronicity. It would also be fine to report if cells exhibited heterogeneity in the dynamics of this transition (in terms of timing and speed), in which case the authors should amend their conclusions here.

We apologize for the confusion. “Synchronization” and “gradual” are both relative terms. We discarded these claims from the manuscript and switched from relative terms to continuous measurements reporting the distributions of the predicted onset of the transitions in a new Appendix Fig S9, and the overall differentiation timing (which appeared in Fig 2H), and amend the conclusions as the reviewer proposes.

Page 9 lines #207-213: *“Visualizing single-cell trajectories showed that most trajectories followed a gradual increase in their differentiation scores (Fig 2G). Measuring the distribution of the per-cell differentiation scores’ temporal derivative (Appendix Fig S8A-B), their integration over time (Appendix Fig S8C-F), the predicted onset Appendix Fig S9), and the predicted duration (Fig 2H) of the differentiation process, suggested that the progression in single-cell differentiation is highly heterogeneous (Fig 2H). These results suggest a heterogeneous gradual transition from an undifferentiated to a differentiated state within a typical timeframe.”*

New Appendix Figure S9:

I also do not think that the validation experiment in Fig 2I is appropriate to test if the transition is gradual. These results are consistent with the time frame over which the transitions are predicted by the model to occur (8.5-14.5h), but these results could also be obtained if cells abruptly transitioned at different time points.

We agree with the reviewer. Gradual versus abrupt transitions are relative terms and thus we decided to discard them from the manuscript. We no longer claim that the results presented in old figure Fig 2I validate the model's prediction and have moved them to Fig 1B describing the experimental setup with respect to the current state-of-the-art.

The authors also claim that simple single-cell measurements are insufficient to quantify cell state transitions (Fig 3). Why not fold these measurements into the model? Couldn't the feature extraction of motility/actin dynamics over specific time windows be combined with these simple measures at timepoints in the middle of these windows? In fact, why present the actin and motility models separately? Wouldn't combining all of these measurements produce an even more accurate model?

The novelty of our manuscript stems from using the model's confidence as a continuous measurement for cell state transitions, rather than optimizing the best model, which we also now explicitly stated in the manuscript (see more in response to reviewer #2). Nevertheless, we implemented this suggestion and trained a new model and reported its superiority in Fig EV3 (see response to reviewer #2).

In the final experiment to uncouple differentiation from fusion, the authors claim that their model can predict differentiation even when fusion is inhibited using a p38 inhibitor in combination with their ERK inhibitor. There are clearly differences in the dynamics of MyoG gain (Fig 5C) and MyoD loss (Fig 5D). In addition, the actin model performs worse in the p38i/ERKi condition; however, it is difficult to assess if the differences in these curves is significant since the authors do not show variance. Even so, the actin model maxes out at a model certainty value of 0.5, which by definition cannot distinguish undifferentiated from differentiated cells.

We revised the figure to show the distributions and assess the statistical significance of these curves.

New Appendix Figure S14:

We acknowledge that the actin model's certainty value maxes out at 0.5 and, by definition, cannot distinguish undifferentiated from differentiated cells. This is a fantastic catch by the reviewer that pointed us to explore this deviation and conclude that the actin machinery was altered. The actin model predicts a monotonically increasing trend, suggesting that the cells are becoming more differentiated over time, and thus can still measure the change of cell state. To investigate whether the differentiation process was altered (as predicted by the actin classifier), or not (as predicted by the motility classifier), we acquired mass spectrometry data that confirmed the differentiation and highlighted the molecular-level differences between the conditions, where one of these changes was an alteration in actin-regulatory proteins. These results suggest that the alteration of the actin model was indicative of an alteration in the actin machinery, most likely associated with lack of fusion and sacromere formation, which are both dependent on actin. These results are now reported in a new Figure 6 (see response to reviewer #2), and demonstrate that our model could identify, *in silico*, subtle changes that can then be verified experimentally.

Summary of main revisions:

1. To test whether the differentiation was altered or not in response to joint inhibition of ERK and P38, we acquired mass spectrometry data to validate the differentiation and identify differences at the molecular level (reviewer #2 and #3). Results are reported in a new Fig. 6.
2. We identified persistent motility as a functional marker for the intermediate states of myoblast differentiation demonstrating, for the first time, that the dynamic single cell state readout can be correlated to other, independently measured, cell readouts (reviewer #2). These results are reported in a new Fig EV4.
3. We showed that the temporal derivative in single cell actin intensity over time could partially distinguish between differentiated and undifferentiated cells but cannot provide a continuous readout of the differentiation state in Fig 3D and in a revised Fig 3F-G (reviewer #1).
4. We analyzed the temporal change in the single cells' differentiation score, its integration over time, and its statistical significance in a new Appendix Fig S8 (reviewer #2).
5. We combine the actin and motility time series, trained a new model and showed that its performance is superior to models trained with actin or motility features (reviewers #2 and #3). Results are reported in a new Fig EV3.
6. We characterized the cells where the actin-based and motility-based models disagree (reviewer #2). Results are reported in a new Appendix Fig S10.
7. We presented the change in actin/speed at the single cell level as a motivation for using these measures for our predictive model (reviewer #2). Results are reported in a new Appendix Fig S1.
8. We plotted the change in the single cells' differentiation score in comparison to the differentiation score at the onset of the experiment to better characterize its "continuity" and provide statistical analysis (reviewer #2). Results are reported in a new Appendix Fig S8.
9. We report the consistency between independent rounds of training in new Appendix Fig S7 (reviewer #2).
10. We report the distributions of the predicted onset of the undifferentiated-to-differentiated transitions as another indication of heterogeneity (reviewer #3). Results are reported in a new Appendix Fig S9.
11. New visualization of the experiment in a new Fig. EV1.
12. Source code and example data are publicly available, <https://github.com/amitshakarchy/muscle-formation-regeneration> (reviewer #1)
13. Extensive clarifications and revisions of the text.

21st Nov 2023

Manuscript Number: MSB-2023-11631R

Title: Machine learning inference of continuous cell state transitions during myoblast differentiation

Dear Assaf,

Thank you for sending us your revised manuscript. We have now heard back from the three reviewers who were asked to evaluate your revised study. As you will see below, reviewers #1 and #2 are satisfied with the performed revisions and support publication. Reviewer #3 is still concerned about the ability of the model to predict new biology. Given that these remaining concerns do not seem to be shared by the other reviewers, we have decided to proceed with publishing the study. We think that addressing the remaining issues raised by reviewer #3 by providing some further clarifications and performing text changes seems sufficient. Reviewer #1 also lists a minor issue that can also be addressed in the revision.

We would also ask you to address some remaining editorial issues listed below.

- Our data editors have noted the following items that need to be corrected in the figure legends:

1. Please indicate the statistical test used for data analysis in the legends of figures 2f; 4b-d; 5b; EV4b-c.
2. Please define the annotated p values *** in the legend of figure 5b as appropriate."
3. Please note that information related to n is missing in the legends of figures 1b; 3g; 4b; 5b-c; 6c.
4. Please note that the error bars are not defined in the legends of figures 1b; 5b-c.
5. Please note that the box plots need to be defined in terms of minima, maxima, center, bounds of box and whiskers, and percentile in the legend of figure 3g.

- The funding information provided in the manuscript text needs to match the information entered in the online submission system. Currently ERC-PoC-2022 SuperFusion is missing from the submission system.

- Please include 5 keywords in the main text.

- DOIs should be removed from the Reference list.

- "Software and data availability" should be renamed to "Data Availability". Please indicate in the Data Availability section where the mass spectrometry data have been made available (e.g. at PRIDE) and include the accession number and link to the data.

- The COI statement needs to be renamed to "Disclosure and Competing Interests Statement".

- Please remove the 'Authors Contributions' from the manuscript. The 'Author Contributions' section is replaced by the CRediT contributor roles taxonomy to specify the contributions of each author in the journal submission system. Please use the free text box in the 'author information' section of the online submission system to provide more detailed descriptions if needed (e.g., 'X provided intracellular Ca⁺⁺ measurements in fig Y').

- The Appendix Tables of Contents is duplicated (it is provided on page 1 and page 17), please correct.

- Please provide a "standfirst text" summarizing the study in one or two sentences (approximately 250 characters), three to four "bullet points" highlighting the main findings and a "synopsis image" (550px width and max 400px height, jpeg format) to highlight the paper on our homepage.

- The movie files should be renamed to Movie EV1-EV3 with the corresponding callouts in the text. For each movie, please provide a short description as a README.txt file, zipped together with the movie file.

- All figure legends should be provided after the References (first the main figure legends, followed by the EV figure legends).

Please resubmit your revised manuscript online, with a covering letter listing amendments and responses to each point raised by the referees. Please resubmit the paper **within one month** and ideally as soon as possible. If we do not receive the revised manuscript within this time period, the file might be closed and any subsequent resubmission would be treated as a new manuscript. Please use the Manuscript Number (above) in all correspondence.

Click on the link below to submit your revised paper.

Kind regards,

Maria

Maria Polychronidou, PhD
Senior Editor
Molecular Systems Biology

If you do choose to resubmit, please click on the link below to submit the revision online before 21st Dec 2023.

IMPORTANT:

Please note that corresponding authors are required to supply an ORCID ID for their name upon submission of a revised manuscript (EMBO Press signed a joint statement to encourage ORCID adoption).

(<https://www.embopress.org/page/journal/17444292/authorguide#editorialprocess>)

Currently, our records indicate that the ORCID for your account is 0000-0002-1477-5478.

Link Not Available

*** PLEASE NOTE *** As part of the EMBO Press transparent editorial process initiative (see our Editorial at <https://dx.doi.org/10.1038/msb.2010.72> , Molecular Systems Biology will publish online a Review Process File to accompany accepted manuscripts. When preparing your letter of response, please be aware that in the event of acceptance, your cover letter/point-by-point document will be included as part of this File, which will be available to the scientific community. More information about this initiative is available in our Instructions to Authors. If you have any questions about this initiative, please contact the editorial office (msb@embo.org).

Reviewer #1:

The revised version addressed all of my concerns and the addition of MS results strengthened the conclusions, thus making it suitable for publication in MSB.

Minor point: the first sentence of the abstract should likely read: "Cells modify their internal..."

Reviewer #2:

The authors have responded comprehensively and insightfully to the concerns raised in the initial review. I appreciate the efforts undertaken to address these points, shedding further light on the conceptual and methodological foundations of the study.

Specifically, the authors have elaborated on the rationale for selecting motility and actin intensity as features for the machine learning model. They have provided an explicit presentation and motivation for using these measures at the single-cell level. The clarification on the integration of temporal information, as evident in the revised manuscript, enhances the understanding of the model's rationale.

Acknowledging the challenge of validating the differentiation score at the single-cell level in the absence of traditional ground truth markers, the authors have taken steps to demonstrate the reliability of their approach, providing indirect evidence. Notably, they showcased a high correlation between time and the differentiation score for both motility- and actin-based classifiers at the single-cell level. Furthermore, they demonstrated the association between the motility-based model and persistent cell migration, establishing a link between the single-cell differentiation score and a functional readout. This emphasizes the feasibility of their proposed approach at the single-cell level.

The authors effectively addressed scenarios where the models disagreed and leveraged the single-cell differentiation score to

make specific predictions about alterations in actin regulators. The subsequent confirmation of these predictions through mass spectrometry analysis underscores the predictive power of their approach, showcasing the utility of their proposed differentiation score.

In conclusion, the authors' responses have significantly strengthened the manuscript by providing additional insights, clarifications, and supporting evidence. Their computational method is now better positioned for publication, demonstrating its utility and potential impact in understanding cell differentiation processes at the single-cell level.

Reviewer #3:

While I think that using a confidence score from a trained model to predict the transition between two states is interesting and potentially powerful for certain applications, I am not convinced of the robustness of the model presented in this manuscript, nor the power of it to generate novel biological insights. I believe this paper is a better fit for a more technical journal.

Several of my previous criticisms highlighted weaknesses associated with the claims made about the behavior and heterogeneity among single cells, and the ability of this model to describe the differentiation characteristics of individual cells. This is a model trained on single-cell data that proposes to follow a single cell phenotype - differentiation - and the real power and promise of an approach like the one presented by the authors is to better understand this process at the single cell level. Instead of clarifying or providing clear data to support claims about the characteristics of single-cell differentiation trajectories, in many cases these data or claims were simply removed. I think that this is important because this heterogeneity is the source of the notable variance in both the actin and motility differentiation scores presented throughout this paper and understanding it (and confronting it properly) may have improved this paper beyond the threshold for publication.

This is particularly problematic in the crucial experiments where the authors seek to use their model to predict biology (Fig 6). In this case in particular, the omission of variance from these plots and relegation of them to Fig S14 seems like a deliberate choice to hide this variance from the reader. Although the authors include some measure of statistical significance in Fig S14, there is not a sufficient description of the significance test used in either the legend or the methods. Independent t-tests at every datapoint along these traces is not an appropriate assessment of confidence and the authors should use a test designed for time-series data (e.g. <https://www.sciencedirect.com/science/article/pii/S0966636218304971>) and indicate so, if they already have.

The 7.5-14.5h differentiation windows suggested by the model is not validated with experimental data. A time-course experiment showing that key differentiation markers change the most during this interval would support this assertion. Global measurement using RNAseq or mass spec would also suffice.

Another biological insight proposed by the authors is the ~3h delay between terminal differentiation and fusion. However, if I understand correctly, their model was trained on data in which the differentiated state was defined as 2.5h prior to fusion, creating a circular logic:

"Hence, we defined the cultures as differentiated for classification 2.5 hours before the first fusion event was observed in the field of view"

Thus, it is not surprising that they find a ~3h gap between terminal differentiation identified by their models and fusion.

Therefore, despite the changes made by the authors, I do not believe that this manuscript possesses the rigor or impact to warrant publication in MSB.

אוניברסיטת בן-גוריון בנגב
Ben-Gurion University of the Negev

Assaf Zaritsky, Ph.D.
Assistant Professor

Faculty of Engineering Sciences
Software and Information Systems Engineering

December 18th, 2023

To: Dr. Maria Polychronidou, Senior Editor, *Molecular Systems Biology*

Point-by-point response

We appreciate the time and constructive feedback provided by all the reviewers during the full process. We are sorry that Reviewer #3's interest in our study was decreased following our revision. We feel that this is due to a misunderstanding and provided clarifications in our response.

Reviewer #1:

The revised version addressed all of my concerns and the addition of MS results strengthened the conclusions, thus making it suitable for publication in MSB.

Thank you.

Minor point: the first sentence of the abstract should likely read: "Cells _modify_ their internal..."

Done.

Reviewer #2:

The authors have responded comprehensively and insightfully to the concerns raised in the initial review. I appreciate the efforts undertaken to address these points, shedding further light on the conceptual and methodological foundations of the study.

Specifically, the authors have elaborated on the rationale for selecting motility and actin intensity as features for the machine learning model. They have provided an explicit presentation and motivation for using these measures at the single-cell level. The clarification on the integration of temporal information, as evident in the revised manuscript, enhances the understanding of the model's rationale.

Acknowledging the challenge of validating the differentiation score at the single-cell level in the absence of traditional ground truth markers, the authors have taken steps to demonstrate the reliability of their approach, providing indirect evidence. Notably, they showcased a high correlation between time and the differentiation score for both motility- and actin-based classifiers at the single-cell level. Furthermore, they demonstrated the association between the motility-based model and persistent cell migration, establishing a link between the single-cell differentiation score and a functional readout. This emphasizes the feasibility of their proposed approach at the single-cell level.

The authors effectively addressed scenarios where the models disagreed and leveraged the single-cell differentiation score to make specific predictions about alterations in actin regulators. The subsequent confirmation of these predictions through mass spectrometry analysis underscores the predictive power of their approach, showcasing the utility of their proposed differentiation score.

In conclusion, the authors' responses have significantly strengthened the manuscript by providing additional insights, clarifications, and supporting evidence. Their computational method is now better positioned for publication, demonstrating its utility and potential impact in understanding cell differentiation processes at the single-cell level.

Thank you.

Reviewer #3:

While I think that using a confidence score from a trained model to predict the transition between two states is interesting and potentially powerful for certain applications, I am not convinced of the robustness of the model presented in this manuscript, nor the power of it to generate novel biological insights. I believe this paper is a better fit for a more technical journal.

Several of my previous criticisms highlighted weaknesses associated with the claims made about the behavior and heterogeneity among single cells, and the ability of this model to describe the differentiation characteristics of individual cells. This is a model trained on single-cell data that proposes to follow a single cell phenotype - differentiation - and the real power and promise of an approach like the one presented by the authors is to better understand this process at the single cell level. Instead of clarifying or providing clear data to support claims about the characteristics of single-cell differentiation trajectories, in many cases these data or claims were simply removed. I think that this is important because this heterogeneity is the source of the notable variance in both the actin and motility differentiation scores presented throughout this paper and understanding it (and confronting it properly) may have improved this paper beyond the threshold for publication.

In our rebuttal response to this reviewer during the previous round of revision, we toned down or removed some of our claims. Specifically, these included:

1. The claim that myoblast differentiation is difficult to study because of the complex heterogeneity. Heterogeneity, in this context, was referred to in the timescales of hours (ERK inhibition) versus days (untreated cells), as reported in our previous study (PMID: 34932950). This is not related to the cell heterogeneity upon ERK inhibition that was what the reviewer was referring to.
2. A similar mistake was our use of two potential mechanisms of (1) synchronized gradual transitions of single-cells, or (2) unsynchronized abrupt transitions. This was the same issue that related to time scales. How could one measure gradual transitions versus abrupt transitions? We tried to relate to this in Fig. 2F where we reported the distribution of the correlations between differentiation score and time, and in Fig. 2H where we reported the distribution of the differentiation process duration. This still does not fully answer the question because “abrupt” is a time-scale dependent definition and so we decided to exclude these claims.

We did not intend to avoid/ignore the reviewer’s comments regarding cell-cell heterogeneity, and in fact, we comprehensively confronted these concerns in response to reviewers #1 and #2, as we detailed below. We apologize for this misunderstanding caused by not mentioning these revisions properly in the context of reviewer #3’s concerns.

1. We identified persistent motility as a functional marker for the intermediate states of myoblast differentiation, demonstrating, for the first time, that the dynamic single-cell state readout can be correlated to other, independently measured, cell readouts

- (reviewer #2). These results explore cell heterogeneity by examining the population distribution of these correlations. Results are reported in a new Fig EV4.
2. We characterized the cells where the actin-based and motility-based models disagree (reviewer #2). These results explore cell heterogeneity by analyzing sub-groups of cells according to the models' agreement. Results are reported in a new Appendix Fig S10.
 3. We reported the distributions of the predicted onset of the undifferentiated-to-differentiated transitions as another indication of heterogeneity (reviewer #3). These results explore cell heterogeneity in the context of onset differentiation time. Results are reported in a new Appendix Fig S9.

This is particularly problematic in the crucial experiments where the authors seek to use their model to predict biology (Fig 6). In this case in particular, the omission of variance from these plots and relegation of them to Fig S14 seems like a deliberate choice to hide this variance from the reader.

We did not intend to hide this variance, in fact, we included this plot explicitly in our response, referring to Fig. S14. We thought that the plot without the variance is clearer to the readers, but in order to avoid this misconception we decided to show the variance in Fig. 5D-E.

Revised Figure 5:

Although the authors include some measure of statistical significance in Fig S14, there is not a sufficient description of the significance test used in either the legend or the methods. Independent t-tests at every datapoint along these traces is not an appropriate assessment of confidence and the authors should use a test designed for time-series data (e.g. <https://www.sciencedirect.com/science/article/pii/S0966636218304971>) and indicate so, if they already have.

We now report a Random Field Theory (RFT)-Based Inference statistical test designed for time-series data in Appendix Figures S4E-F, S8H-I, and S15.

Revised Appendix Figure S4:

Revised Appendix Figure S8:

New Appendix Figure S15:

The 7.5-14.5h differentiation windows suggested by the model is not validated with experimental data. A time-course experiment showing that key differentiation markers change the most during this interval would support this assertion. Global measurement using RNAseq or mass spec would also suffice.

To the best of our knowledge, there is currently no molecular determinant that is temporally correlated to the process of differentiation. The existing markers are “binary”, meaning that if they are expressed, the cell is considered differentiated, as we show for MyoG in Figure 1. The increase in the number of MyoG-expressing cells is correlated with the differentiation time course predicted by the model. Moreover, the existing differentiation markers shuttle in and out of the nucleus as differentiation progresses rather than change their level of expression, which precludes RNAseq as a validation method.

Another biological insight proposed by the authors is the ~3h delay between terminal differentiation and fusion. However, if I understand correctly, their model was trained on data in which the differentiated state was defined as 2.5h prior to fusion, creating a circular logic:

"Hence, we defined the cultures as differentiated for classification 2.5 hours before the first fusion event was observed in the field of view"

Thus, it is not surprising that they find a ~3h gap between terminal differentiation identified by their models and fusion.

The differentiation time was defined as 2.5 hours before the first fusion event. Note the heterogeneity in fusion timing in Fig. 4B (see below) that spans over ~10 hours. In fact, the median timing of the fusion events occurs ~3 hours after the first fusion event occurred. Also, the differentiation time is determined computationally according to a thresholded differentiation score. Thus, the mean of ~3h gap between predicted terminal differentiation and fusion is not trivially derived from the definition of differentiation timing during training.

We added this clarification to the manuscript in page #12 lines #297-300: *“The mean of a ~3 hour gap between predicted terminal differentiation and fusion is not trivially derived from the definition of differentiation timing during training because (1) the differentiation time at training was defined as 2.5 hours before the first fusion event, and (2) the heterogeneity in fusion timing spans over ~10 hours (Fig. 4B).”*

Therefore, despite the changes made by the authors, I do not believe that this manuscript possesses the rigor or impact to warrant publication in MSB.

We are sorry that our revisions reduced your enthusiasm about our manuscript, and hope that these clarifications help.

4th Jan 2024

Manuscript number: MSB-2023-11631RR

Title: Machine learning inference of continuous cell state transitions during myoblast differentiation

Dear Assaf and Ori,

Thank you again for sending us your revised manuscript. We are now satisfied with the modifications made and I am pleased to inform you that your paper has been accepted for publication.

Best wishes,

Maria

Maria Polychronidou, PhD
Senior Editor
Molecular Systems Biology
